Comment

# A call for a unified and multimodal definition of cellular identity in the enteric nervous system

Homa Majd [ID] [1,2,7 ✉], Andrius Cesiulis[1,2,7], Ryan M Samuel[1,2,7], Mikayla N Richter[1,2], Nicholas Elder[1,2], Kwun Wah Wen[3], Richard A Guyer[4], Marlene M Hao [ID] [5], Lincon A Stamp [ID] [5], Allan M Goldstein[4] & Faranak Fattahi [ID] [1,2,6 ✉]

The enteric nervous system (ENS), the largest division of autonomic nervous system, is a tantalizing frontier in neuroscience. With the advent of single-cell transcriptomics, the ENS has been increasingly well-characterized. Precise functional mapping of enteric neuron diversity is critical for understanding ENS biology and disease, but technical barriers remain. We used different approaches to compare and contrast functional annotations of several independently-reported ENS datasets. Differential module scoring, co-expression and correlation analysis, unbiased biological function hierarchical clustering, data integration and label transfer highlighted substantial discrepancies stemming from an overreliance on transcriptomics data without adequate tissue validations. For understanding enteric neurons' functional identity, it is imperative to expand tissue sources and incorporate technologies such as multiplexed imaging, electrophysiology, spatial transcriptomics, as well as comprehensive epigenome, proteome, and metabolome profiling. Harnessing human pluripotent stem cell models provides unique opportunities for ENS lineage tracing and offers unparalleled scalability and amenability to genetic and functional screens. We encourage a paradigm shift in our comprehension of ENS cellular and functional complexity by calling for large-scale collaborations and research investments.

**Keywords** Neurochemical Coding; Comparative Analysis; Neuronal Classification

As the largest and most complex division of the autonomic nervous system, the enteric nervous system (ENS), holds the utmost significance in normal physiology and pathophysiology of the gut and disorders of gut-brain interaction (DGBIs). Comprising the cell bodies and projections of over 500 million neurons, and many more glia, the ENS is an intricate network that spans the length of the gastrointestinal (GI) tract. The ENS autonomously manages gut motility, secretion, absorption, local blood flow, and barrier function while also communicating with the brain, neuroendocrine and immune systems, and the microbiome. To fully understand the capabilities of the ENS, it is crucial to attain a comprehensive understanding of the molecular and functional diversity of its cell types. By unraveling the complex molecular and cellular processes, including epigenetic, transcriptional, post-transcriptional, translational, and post-translational regulatory mechanisms, as well as the context in which the ENS operates, such as the gut microenvironment, metabolic signals, and microbial and immune responses, we can begin to appreciate the remarkable autonomy and adaptability of the ENS in managing the diverse functions of the GI tract.

Towards this goal, ensuring accurate classification and characterization of enteric neurons is a critical first step with high-stakes implications for understanding the biology of the ENS and its associated diseases. Well-characterized cell populations and their markers serve as the basis for establishing in vitro and in vivo models, enabling researchers to explore ENS development, function, and crosstalk with other organ systems and external factors. Furthermore, consistent cellular annotations are crucial for scientific communication and the interpretation of emerging research findings.

In recent years, the ability to model the ENS using human stem cell-derived in vitro systems has opened new avenues for regenerative therapies and drug discovery. However, the utility of these models hinges on their fidelity—they must closely mimic the molecular and functional characteristics of native ENS cells to serve as reliable platforms for disease modeling, therapeutic screening, and cell replacement strategies. This requires not only robust differentiation protocols but also rigorous benchmarking against primary human tissue.

To establish such benchmarks, we turned to single-cell transcriptomic datasets as a means to comprehensively compare in vitro-derived neurons to their in vivo counterparts. Yet, this effort revealed a critical barrier: there is no consistent ground truth

[1]Department of Cellular and Molecular Pharmacology, University of California, San Francisco, San Francisco, CA 94158, USA. [2]Eli and Edythe Broad Center of Regeneration Medicine and Stem Cell Research, University of California, San Francisco, San Francisco, CA 94143, USA. [3]Department of Pathology, University of California, San Francisco, San Francisco, CA, USA. [4]Department of Pediatric Surgery, Massachusetts General Hospital, Boston, MA, USA. [5]Department of Anatomy and Physiology, the University of Melbourne, Parkville, VIC, Australia. [6]Program in Craniofacial Biology, University of California, San Francisco, San Francisco, CA, USA. [7]These authors contributed equally: Homa Majd, Andrius Cesiulis, Ryan M Samuel. ✉E-mail: Homa.Majd@ucsf.edu; Faranak.Fattahi@ucsf.edu
https://doi.org/10.1038/s44318-025-00559-1 | Published online: 15 September 2025

for enteric neuron identity in human tissue. Instead, we found significant discrepancies across published datasets in how enteric neurons are classified and annotated, undermining our ability to assess model fidelity and raising broader concerns about standardizing ENS characterization.

Historically, enteric neuron subtypes have been defined by morphology, neurochemistry, and electrophysiology, primarily through studies in animal models such as guinea pig, mouse, and rat (Furness, 2000). These models have been foundational in describing major neuron classes, including intrinsic primary afferent neurons (IPANs), interneurons, motor neurons, and secretomotor neurons, with additional subtype diversity (Furness, 2006; Mann et al, 1995; Brehmer, 2021). Despite the broad neurochemical landscape of the ENS, a limited set of markers, such as NOS1 for inhibitory motor neurons, CHAT for excitatory motor neurons, and CALCA/B for IPANs, are routinely used to identify these populations (Qu et al, 2008).

While it is impressive how large swaths of ENS function are coordinated by these neurotransmitters and neuropeptides, it also raises the question of whether yet-to-be-determined factors confer cell-type-specific functional properties (Wallace and Sabatini, 2023). These features might include secreted peptides and proteins, electrophysiological features mediated by ion channels and transporters, metabolic profiles, and specialized capabilities for cell-cell interaction or synapse formation. To accurately understand the identity of individual cell types, it is also crucial to consider the contextual information in which they operate, including their position in the gut. Histochemical and functional assays have identified differences along the length of the GI tract, as well as between species, highlighting the need for more systematic efforts towards comprehensive mapping and profiling of the ENS (Furness, 2000).

This study was driven by the need for accurate, standardized molecular definitions of human enteric neurons to enable future advances in basic understanding of ENS biology and development of hPSC-derived models for therapeutic use. Here, we systematically examine existing molecular profiles of enteric neurons, identify key sources of annotation inconsistency, and propose a framework to guide more consistent and reliable classification moving forward.

## Single-cell transcriptomics identifies distinct enteric neuron subtypes

With the emergence of single cell and single nuclei transcriptomics (scRNA-seq and snRNA-seq) over the past decade, enteric neurobiologists have begun to overcome the gap posed by the limited availability of histochemical and functional probes to distinguish between subtypes of enteric neurons. In fact, several scRNA-seq datasets of the ENS have been recently published and reviewed, including mouse, human, and human pluripotent stem cell (hPSC) derived enteric neurons (Drokhlyansky et al, 2020a; Wright et al, 2021; May-Zhang et al, 2021; Majd et al, 2025; Morarach et al, 2021b; Elmentaite et al, 2021b; Richter et al, 2023; Dharshika and Gulbransen, 2023; Guyer et al, 2022). The datasets of primary mouse and human intestine published by Ulrika Marklund (UM-mouse (Morarach et al, 2021b), Data ref: Morarach et al, 2021a, Fig. EV1A), Aviv Regev (AR-mouse and AR-human (Drokhlyansky et al, 2020a), Data ref: Drokhlyansky et al, 2020b, Fig. EV1B, C), Sarah Teichmann (ST-human (Elmentaite et al, 2021b), Data ref: Elmentaite et al, 2021a, Fig. EV1D) and colleagues provide immensely valuable transcriptional profiling of a highly intricate biological tissue that is inherently challenging to acquire.

These studies present functional annotations for clusters of neurons, with varying numbers and identities of functional classes and subclasses (Fig. 1A). For instance, the term IPAN is missing in annotations used in AR-human and AR-mouse, and PSN (putative sensory neurons) is used instead. Each dataset resolves different numbers of IMN clusters with UM-mouse containing two clusters, AR-mouse containing seven, ST-human containing only one, and AR-human containing five (Fig. 1A). Interestingly, none of these datasets have detected, annotated, or mentioned intestinofugal enteric neurons, afferent neurons that project to and form synapses with sympathetic ganglia (Furness, 2006; Mann et al, 1995). Cart, expressed by gene Cartpt, has been used to mark intestinofugal neurons. Cart in mouse and CARTPT in human was detected across the datasets (Appendix Fig. S1A–D). In UM-mouse, the highest levels of Cart expression was detected in IMN2, EMN4, IPAN2, and IPAN3 (Appendix Fig. S1A). Meanwhile, Cart expression

was highest in AR-mouse PIMN2,3,6,7 and PIN1-3 (Appendix Fig. S1B). ST-human's CARTPT expression was high in both IPAN/IN clusters of branch A (A2 and A3), while CARTPT was primarily detected in PIMN 3–5 and PEMN3 in AR-human (Appendix Fig. S1C,D). While differences may also be due to subjectivity in clustering resolution and the analysis pipeline, the power of primary EN datasets has also been hindered by technical challenges arising from the limited number of cells in specific clusters, particularly in human studies. Furthermore, these datasets encompass primary ENS neurons from various species, distinct developmental stages, different regions of the GI tract, and employ different methodologies. This can be considered both a strength and a caveat. On one hand, it provides valuable insights into the extent of neuronal diversity within the ENS. On the other hand, it presents challenges when attempting to infer the common features that define the identity of specific neuronal subtypes.

## Markers used for the identification of neuronal subtypes in different datasets are inconsistent

Transcriptomics datasets of the ENS have been analyzed in isolation. Consequently, the list of genes used to annotate functional subtypes is different in each paper. Even among shared genes, the expression patterns vary widely across these datasets (Figs. 1B,C and EV1E–R). We examined the complete list of genes used for annotating primary enteric neuron clusters, and found that only three, NOS1, TAC1, and PENK, were shared across the three studies (Fig. 1B). Of the 32 UM-mouse markers (Fig. EV1E), several were not detected in AR-human (Fig. EV1H), and for many others, expression was observed in different neuronal classes across datasets (Fig. EV1H–L). For example, markers enriched in UM-mouse IPANs or EMNs often appeared in AR-human IMNs, PSVNs, or PINs, while markers characteristic of ST-human clusters were either absent in AR-human or expressed in unexpected clusters. Even for shared markers, cell-type annotations differed between studies (Figs. 1C and EV1E–P red boxes, EV1Q–R), with some markers shifting between IPAN, PSN, and EMN identities depending on the dataset. TAC1, a marker

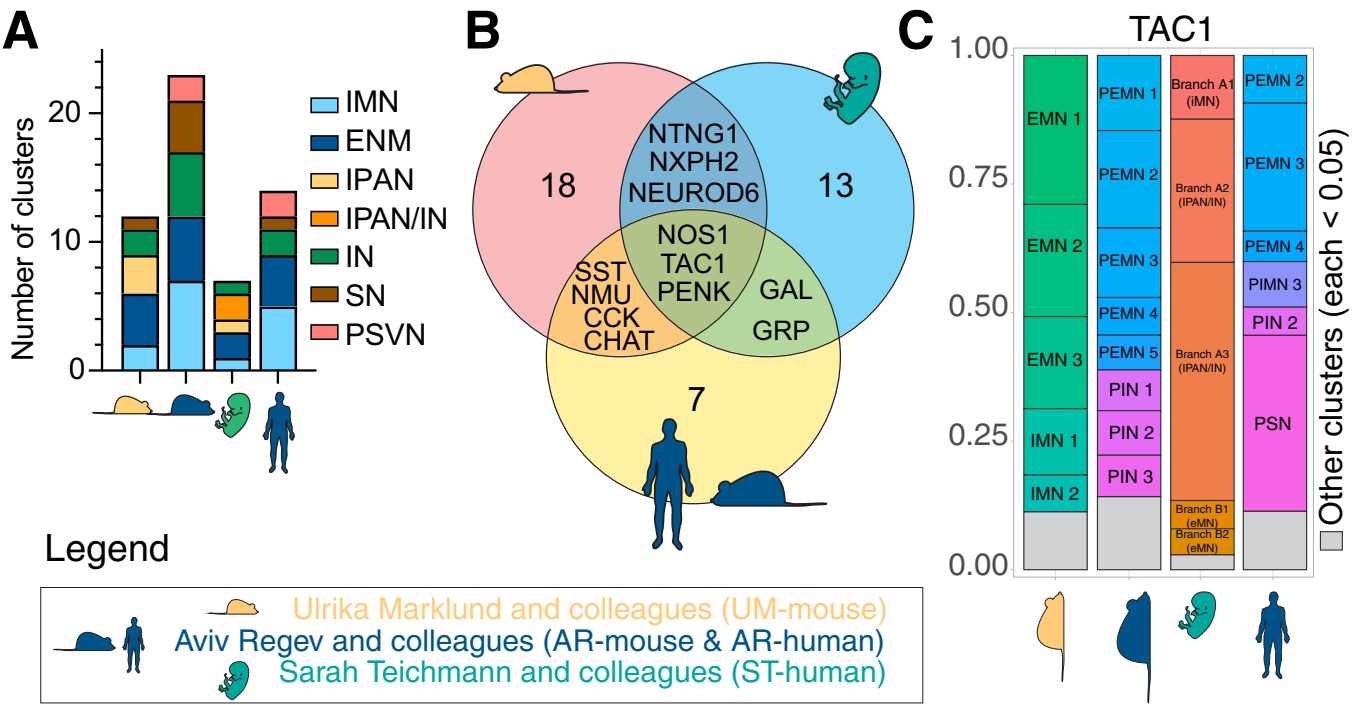

**Figure 1. Cross-dataset expression of primary enteric neuron cluster-specific markers.**

(A) Total number and distribution of enteric neuron cluster annotations Fig. EV1A–D. (B) Venn diagram indicating shared markers used by each study. (C) Bar plot proportion of different functional annotations described for TAC1+ enteric neurons across primary ENS datasets.

attributed to IPAN/IN in ST-human, is a PSN marker in AR-human and an EMN marker in UM-mouse (Fig. 1C).

Our systematic analysis of ENS datasets in parallel highlights substantial discrepancies between cluster-specific transcriptional markers, functional annotations, and the number of subtypes resolved in each dataset. These discrepancies suggest that much of the heterogeneity within the ENS remains to be determined and that annotating functionality based on the expression of only a handful of genes is insufficient.

Enteric neuron identity is often described based on their neurochemical properties, including nitrergic, cholinergic, glutamatergic, catecholaminergic, GABAergic or serotonergic. While such classifications are derived from gene expression data, the functional significance of specific neurotransmitter signaling within the ENS circuitry, particularly for less well-characterized types like glutamatergic neurons, remains to be fully elucidated. This highlights the need for caution when inferring functional roles solely from transcriptional markers. Moreover, it has been shown that enteric neurons can co-express distinct neurotransmitter markers (Qu et al,

2008), which suggests that a single neurotransmitter cannot serve as a specific marker for annotation of transcriptionally distinct enteric neuron subtypes and reaffirms the hypothesis that each neuron can take on multiple neurochemical identities. To further explore this hypothesis at the single-cell resolution, we designed a stringent two-step approach to define an enteric neuron's neurochemical identity (Fig. 2A). In the first step, we identified neurons that expressed the hallmark rate-limiting neurotransmitter synthesis enzymes (for example, NOS1). In the second step, neurons that passed step 1, i.e., expressed the hallmark marker gene, were module scored based on their expression of a curated list of neurotransmitter metabolism enzymes and transport proteins (Appendix Table S1). For example, neurons were annotated as nitrergic if they expressed NOS1 (step 1) and scored highly for NO metabolism and transport genes NOS1AP, ARG1/2, ASL, and ASS1 (step 2). Neurons that passed both steps were binned into a particular class of neurotransmitter identity. This ensured a stricter annotation process, which necessitated the expression and detection of genes essential for the synthesis and release

of the neurotransmitters. As many of the genes related to neurotransmitter synthesis are often transcripts which are produced by the neurons in relatively low abundance, we performed these annotations in parallel based on the authors original published reads (RNA) as well as on imputed read counts to correct for gene-dropout events. To verify that these complex neurochemical and transcriptional identities are physiologically relevant, we applied the same characterization criteria to both primary mouse and human ENS datasets (Drokhlyansky et al, 2020a; Morarach et al, 2021b) (Fig. 2A). We found that most enteric neuron datasets contained neurons from every neurotransmitter identity class (Fig. 2B). We then compared the overall abundance of neurons within each neurochemical class across each dataset irrespective of whether a neuron is predicted to synthesize multiple neurotransmitters (Fig. 2B). By concatenating the individually predicted neurochemical identities, we found that primary enteric neurons contain complex neurochemical identities, where neurons are predicted to synthesize either one, two, three or more neurotransmitters (Fig. 2C,D). We confirmed this in primary

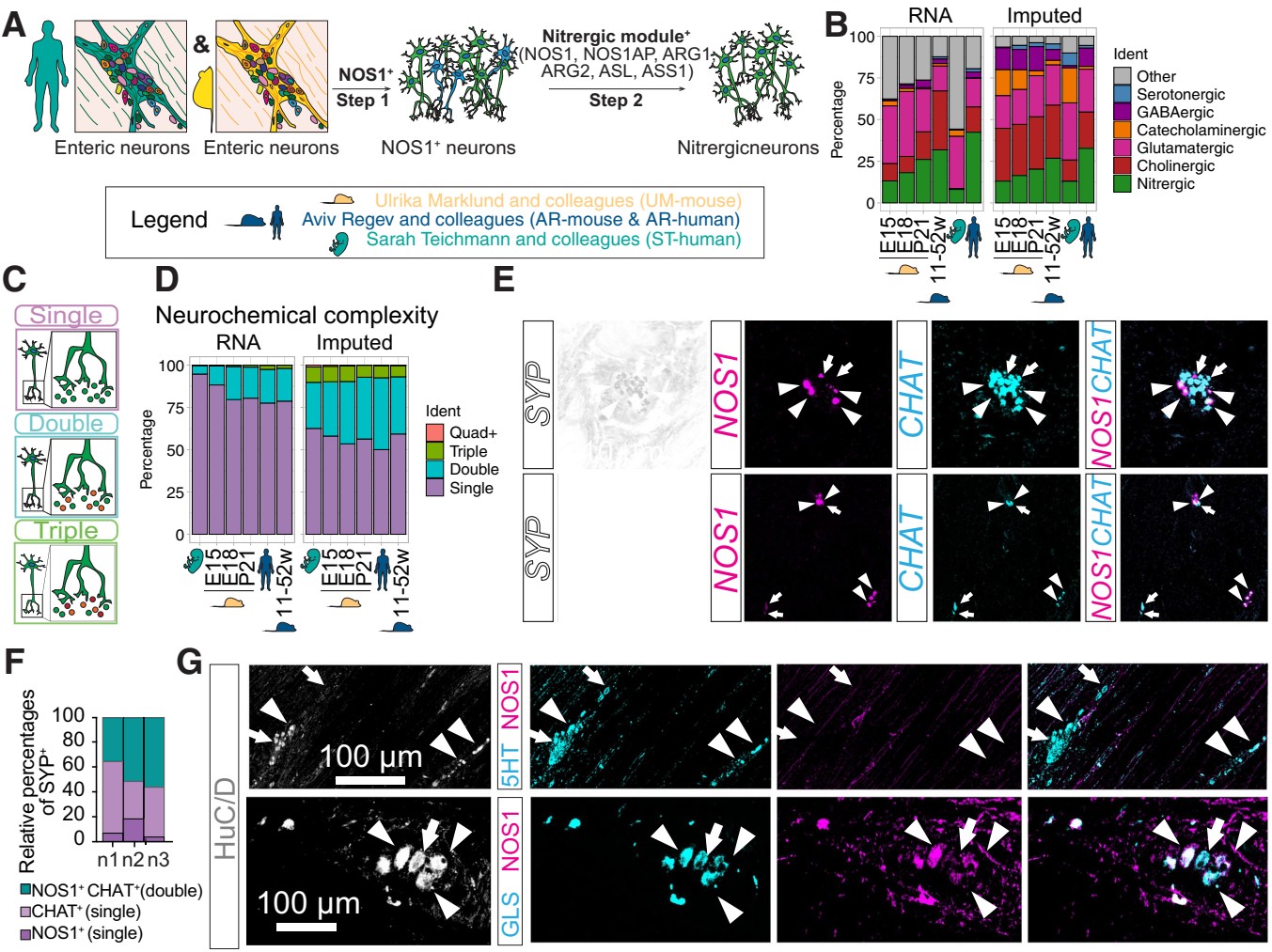

**Figure 2. Comparison of enteric neuron cell types and subtypes in primary enteric datasets.**

(A) Schematic of 2-step neurochemical identity annotation of hPSC-derived enteric neurons. (B) Overall percentage of neurotransmitter-synthesizing neurons in mouse and human primary enteric neurons. (C, D) Schematic (C) and percentage (D) of neurons showing mono-and multi-neurotransmitter profiles in mouse and human primary enteric neurons. (E, F) RNA labeling representative images (E) and quantification (F) analysis of primary human stomach with probes against SYP, NOS1 and CHAT. Arrows indicate exclusive expression and arrowheads indicate colocalized expression of NOS1 and CHAT. (G) Immunostaining of primary human stomach with antibodies against NOS1, 5-HT, and HuC/D (top), and NOS1, GLS, and HuC/D (bottom).

human stomach tissue using in situ hybridization of enteric neurons expressing NOS1 (nitrergic marker) and CHAT (cholinergic marker) transcripts, which confirmed the relative quantifications in snRNA-seq datasets (Figs. 2E,F and EV2A,B). Protein immunostaining of additional neurotransmitter marker genes (Fig. 2G) also showed neurons that were positive for both NOS1 and 5-HT (serotonergic marker) (Fig. 2G top) and NOS1 and GLS (glutamatergic marker) (Fig. 2G bottom). The breakdown of neurons in each single (Fig. EV2A) and double (Fig. EV2B) neurochemical class confirms the presence of similar types of neurons across all

datasets. Finally, we visualized the abundance of each neurochemical class within each author's original cluster assignments, which further highlighted the discrepancies between the utility of using a single neurochemical identity for determining functional neuron annotations in the ENS. As noted previously, transcriptional indications of a neuron producing NO resulted in the fairly consistent assignment of an IMN identity. However, clusters with transcriptional evidence of acetylcholine production accounted for annotations of EMN, IN or IPAN/SN across all datasets (Fig. EV2C). Additionally, these current functional annotations suggest some neurochemical classes

may have species-specific roles, such as catecholamine production being specific to human INs versus mouse SNs (Fig. EV2C).

## Beyond known markers: unbiased detection of neuronal subtypes across different datasets

Since the curated lists of identity genes used in various independent studies did not consistently mark enteric neuron subtypes, we reasoned that unbiased comparison of gene expression signatures could help identify shared and distinct cell populations. There are several categories of computational tools that enable such comparisons in

scRNA-seq datasets (Abdelaal et al, 2019), including differential expression analysis, signature scores, co-expression network analysis, batch effect removal and data integration, annotation/label transfer, and trajectory analysis. Here, we employed some of these strategies to comprehensively compare these enteric neuron datasets.

## Label transfer

Unbiased machine learning–based label transfer methods enable annotation of scRNA-seq and snRNA-seq datasets without manual input (Abdelaal et al, 2019; Pasquini et al, 2021). These methods vary in sensitivity to input features, population size, and performance across datasets, and are broadly classified as supervised or unsupervised. Supervised methods, such as K-nearest neighbors (KNN) and support vector machine (SVM), perform well with a well-annotated, representative reference but may be biased if reference diversity does not match the query. Unsupervised methods, in contrast, do not require annotated references and can manage heterogeneous populations, but may struggle with rare cell types or small datasets. Comparative analyses show most classifiers perform well across datasets, with reduced accuracy in complex datasets with overlapping classes or deep annotations (Abdelaal et al, 2019; Pasquini et al, 2021).

For mapping neuronal subtypes across primary ENS datasets, we used SingleCell-Net (SCN) (Tan and Cahan, 2019), a supervised random forest–based method shown to be accurate, efficient, and robust. SCN trains on random subsamples of a reference, tests performance on remaining cells, and annotates query datasets using the trained model, assigning a 'random' identity if no match exists. Using 100 cells per cluster as the training set, we applied SCN to assess how primary ENS neuron clusters would be annotated under alternative published criteria, relying solely on transcriptional profiles without prior annotation bias (Appendix Table S2).

In SCN comparisons (Figs. 3A–C and EV3A–J), UM-mouse subtypes were most accurately annotated using AR-mouse as the reference (Fig. 3B), though many subtype matches reflected shifts in classification across datasets. Using ST-human as a reference often reassigned UM-mouse clusters to broader categories, with some clusters showing substantial cross-

annotation (Fig. EV3C). When AR-human was used as the reference, nearly all UM-mouse clusters were assigned a PIMN identity except for IPAN1 (Fig. EV3D). In contrast, UM-mouse as reference failed to recover functional neuron identities consistent with ST-human, AR-human, or AR-mouse annotations (Fig. EV3A,G,I). AR-mouse and AR-human comparisons performed best when matched within the same study (Fig. EV3F), though some identities collapsed into single dominant categories, suggesting incomplete capture of biological diversity due to species or technical factors.

SCN also revealed limited transcriptional concordance between human datasets: AR-human PIMN identity was assigned to nearly all ST-human clusters (Fig. EV3B), while ST-human frequently annotated AR-human clusters as IN, even for populations labeled differently in their source dataset (Fig. 3C).

Our unbiased supervised label transfer analyses further confirmed that annotations of neuron subtypes were specific to each dataset and not transferable across the board. These disparities raise intriguing questions regarding the extent to which they stem from limited statistical power, as opposed to meaningful biological differences. In addition, plasticity of the ENS is an important consideration that has been the topic of several recent studies. Alterations in neuronal activity and gene expression patterns through the circadian cycle have been characterized in the murine ENS (Leembruggen et al, 2025; Drokhlyansky et al, 2020a), as well as changes in neuronal identity and function through the estrus cycle (Balasuriya et al, 2021). Overall, the findings underscore the importance of expanding tissue sources from diverse organisms, encompassing various developmental stages, and obtaining larger ENS datasets comprising a broader spectrum of neurons. Furthermore, the results highlight that rethinking annotation strategies, supported by rigorous experimental validations, are essential to advance our understanding of enteric neuron identities and function.

## Module scoring

As a common analysis method, module scoring is used to assess the activity or enrichment of predefined gene sets or modules in individual cells or clusters. By capturing coordinated changes in the expression of hundreds of genes within a

module, module scoring can capture biologically significant relationships even if individual genes do not show strong differential expression. It also offers a tool to reduce the dimensionality of the data by summarizing the expression of gene sets in each cell, simplifying visualization and analysis. However, module scoring can be context-dependent, and the relevance of predefined gene sets might be biased. Here, we used the top 100 differentially expressed genes for each cluster as the subtype signature modules in each dataset.

Module-based analysis reinforced the lack of consistent functional annotations across datasets (Figs. 4A and EV4A–C). Correlations were generally preserved within IMN and within EMN clusters (Fig. EV4A, white boxes), but many modules showed unexpected matches. For example, AR-human PIN modules aligning with UM-mouse IPAN or EMN clusters, and PSVN modules with UM-mouse SN (Fig. EV4A, red boxes). Similar discrepancies emerged when AR-human modules were scored in ST-human clusters, with PIMN and PEMN modules often showing the highest expression in clusters annotated as IN or IPAN (Fig. EV4B, red boxes). Reciprocal analysis of UM-mouse modules in ST-human again revealed cross-annotation, with IMN modules correlating more strongly with IN clusters and EMN modules aligning with IPAN/IN (Fig. EV4B, red boxes). Taken together, our module-based annotations indicate that the functional annotations of ENS neurons is subject to varying interpretation and accuracy when compared across different studies.

## Correlation analysis

Co-expression network analyses identify gene modules or clusters with coordinated expression patterns across samples. Using Spearman correlation, we can compare gene expression profiles between clusters from different datasets to assess potential functional relationships. Spearman is robust to outliers but sensitive to small sample sizes. High positive correlations suggest functional similarity, negative correlations indicate distinct profiles, and values near zero imply little or no relationship.

To compare enteric neuron subtypes in different datasets, we performed Spearman correlation analysis using 3000 or 100 anchor features. Even though the 100-feature analysis showed higher similarity

scores overall, the majority of functional annotations did not align between the clusters of different studies (Fig. EV4D,E). Overall, IMN and EMN clusters showed positive correlation between datasets, with the mouse datasets exhibiting stronger correlation compared to the human datasets, likely due to more accurate annotations as a result of more extensive functional and molecular data in mice. However, the relationship between motor neuron subtypes was not a one-to-one correlation, as anticipated due to the presence of varying subtypes within each motor subtype. Furthermore, both the human-human and mouse-mouse analyses revealed clear

discrepancies between the datasets, with Fig. EV4D,E highlighting consistent (blue boxes) and inconsistent (red boxes) functional annotations. For example, the highest correlation score for the ST-human IN cluster was with AR-human PIMN4, while correlation with AR-human PIN clusters was negative (Fig. EV4D). While the highest correlation of UM-mouse EMN1-3 (0.91, 0.8, and 0.86, respectively) was scored in AR-mouse PEMNs, the highest correlation score of UM-mouse EMN4 was with AR-mouse PIN 2 and 3 (0.86 and 0.83, respectively) (Fig. EV4E).

Our correlation analysis of enteric neuron datasets uncovers substantial gene

expression differences between clusters that are annotated similarly in different datasets. While some positive correlations exist for well-characterized functional subtypes (such as some motor neuron populations), the overall similarities remain modest. These disparities again highlight the insufficiency of transcriptional comparisons to assign functional annotations and the potential revelation of the ENS being more complex and possibly more functionally diverse than previously identified. Thus, the collection and analysis of additional and larger ENS datasets backed by rigorous functional experimental validations is warranted.

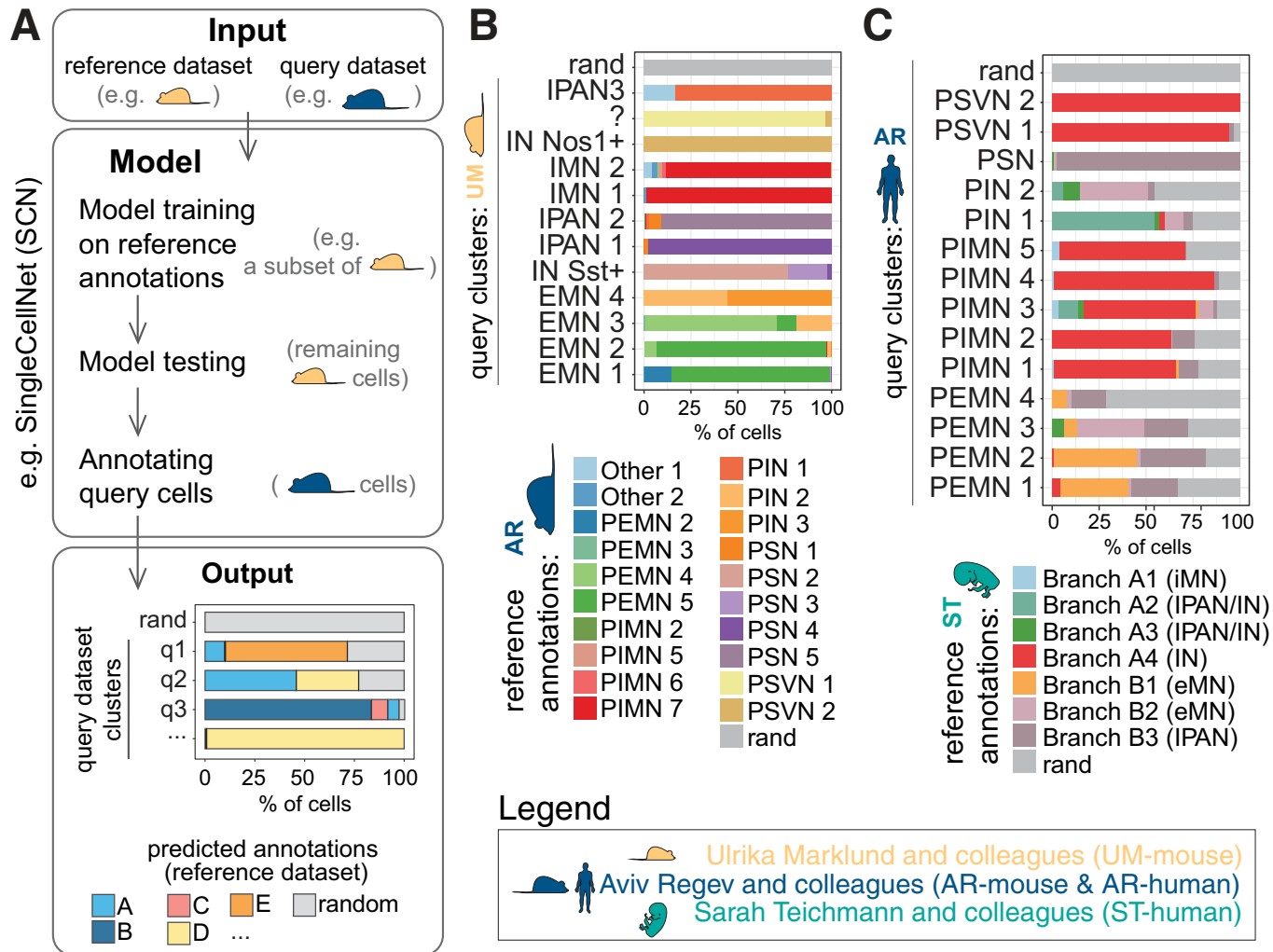

**Figure 3. Unbiased cross-dataset classification of primary enteric neurons using SingleCellNet.**

(A) Schematics of unbiased label transfer using SCN (Tan and Cahan, 2019). (B–C) Reference primary enteric neuron scRNA-seq datasets of AR-mouse (B) and ST-human (C) were used to train SingleCellNet (Tan and Cahan, 2019). These models were then used for label transfer and cross-annotation in the other datasets. Please see Methods for more details. The abbreviations (consistent with the commonly used denotations in the field and in the original papers): IMN (inhibitory motor), EMN (excitatory motor), IN (interneuron), IPAN (intrinsic primary afferent), PSVN (putative secretomotor/vasodilator), and SN (sensory). The inclusion of "P" in the AR-mouse and AR-human indicates "putative" as originally termed.

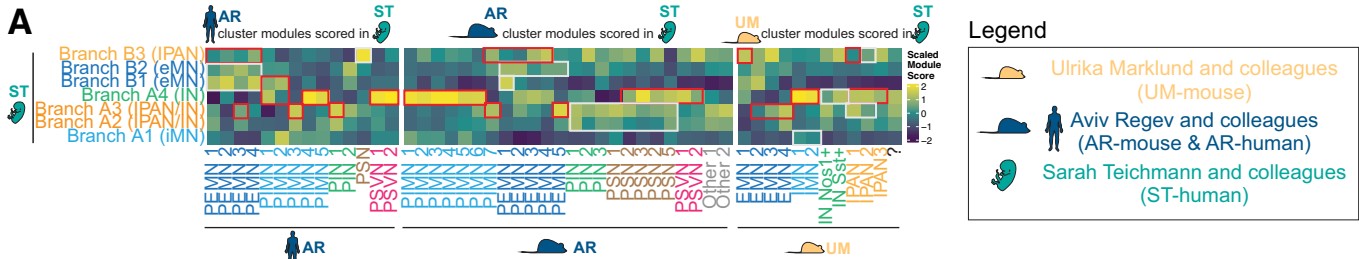

**Figure 4. Cross-dataset module scoring of primary enteric neuron clusters.**

(A) Heatmap of the average module scores of AR-human, UM-mouse, and AR-mouse neuronal subtype transcriptional signatures in ST-human. Please see Methods for more details. Please see Methods for more details.

## Integration

Data integration methods aim to merge multiple datasets to create a unified representation of cell types (Argelaguet et al, 2021). Various integration methods have been developed to align and harmonize datasets into a shared low-dimensional space, enabling comparison across datasets. By incorporating noise reduction and batch correction techniques, integration strategies aim to retain biologically relevant information and preserve the underlying biological relationships between cells. Cells that represent similar gene expression profiles across different datasets tend to group together and enable more accurate downstream analysis such as clustering and differential expression analysis as well as label transfer. Harmony (Korsunsky et al, 2019) is a graph-based integration method that aims to capture the similarities and distances in gene expression profiles of different datasets. The integrated graph resulting from aligning the graph structures of different datasets enables direct comparison and visualization.

Integration of UM-mouse and AR-mouse ENS datasets shows that while some of the Harmony clusters in the integrated UMAP space are composed of cells from both datasets, five of the 14 clusters are almost entirely derived from one dataset, and six other clusters are >75% composed of cells from one dataset (Fig. EV4F–H). Cluster 6 is almost 50% derived from UM-mouse IPAN and 50% AR-mouse PIN. Clusters 7 and 14 show a balanced integration of UM-mouse IPAN and AR-mouse PSN (Fig. EV4F–H). So, while two clusters appear to properly integrate cells with similar functional annotations, this is not the case for the rest of the integrated

dataset, and the datasets largely remain distinct post-integration.

The integration of ST-human and AR-human datasets (Fig. EV4I) revealed distinct differences in the transcription profiles of cells from each dataset. Each of the 13 Harmony clusters were almost entirely derived exclusively from either the ST-human or the AR-human datasets (Fig. EV4J,K). While this clear divide in integration might be attributed to variations in the developmental stages of the neurons, the absence of any integration, even in the well-studied motor neuron clusters, underscores the need for a critical review and comprehensive analysis of the annotations, sample preparation, and development of improved computational and analysis techniques.

Altogether, to gain a deeper understanding of the observed discrepancies and ensure accurate cluster annotations, it is essential to thoroughly assess the experimental procedures, data processing pipelines, and potential batch effects. By addressing these challenges, we can enhance the reliability and validity of the integration results and advance our understanding of neuronal diversity within the ENS.

## Clustering based on gene set enrichment analysis (GSEA)

Hierarchical clustering is a valuable unsupervised clustering strategy to help identify similarities and differences in gene expression patterns among cells or clusters of different datasets. It offers a particularly useful method when assessing the overall relationship and similarities between different populations across multiple datasets, identifying conserved cell types, as well as identifying potentially novel cell

populations without biasing the analysis with prior knowledge. It constructs a tree-like structure (dendrogram) where branches are formed through merging and splitting based on similarity. Like other methods, it is essential to consider batch effects, data normalization, and sample size. Other unbiased clustering methods include K-means clustering, which requires the user to specify the K number of clusters in advance, or graph-based clustering, which leverages the graph structures and identifies clusters based on their relative similarities compared to other cells/clusters.

To assess the functional similarities of primary mouse and human ENS neuron subtypes based on defined gene sets, we conducted gene set enrichment analysis (GSEA) using the gene ontology biological process (GOBP) gene sets. Comparing clusters based on the enrichment of a large number of gene sets offers solutions to many technical challenges when comparing independently produced datasets. Attributing the expression of numerous genes to a single GO pathway allows for the detection of common pathways, even if different subsets of the gene set were detected in different datasets. This overcomes technical limitations attributed to batch effects and gene dropout due to differences in sequencing depths between datasets. Additionally, this method allows for cross-species comparison without the need for homologous gene assumptions, as the same pathways have been annotated for both mouse and human-specific genes and gene family members.

We performed this analysis on the significantly upregulated gene lists of each neuron cluster, which were calculated independently for each dataset and rank-ordered by $\log_2$ fold change. We employed

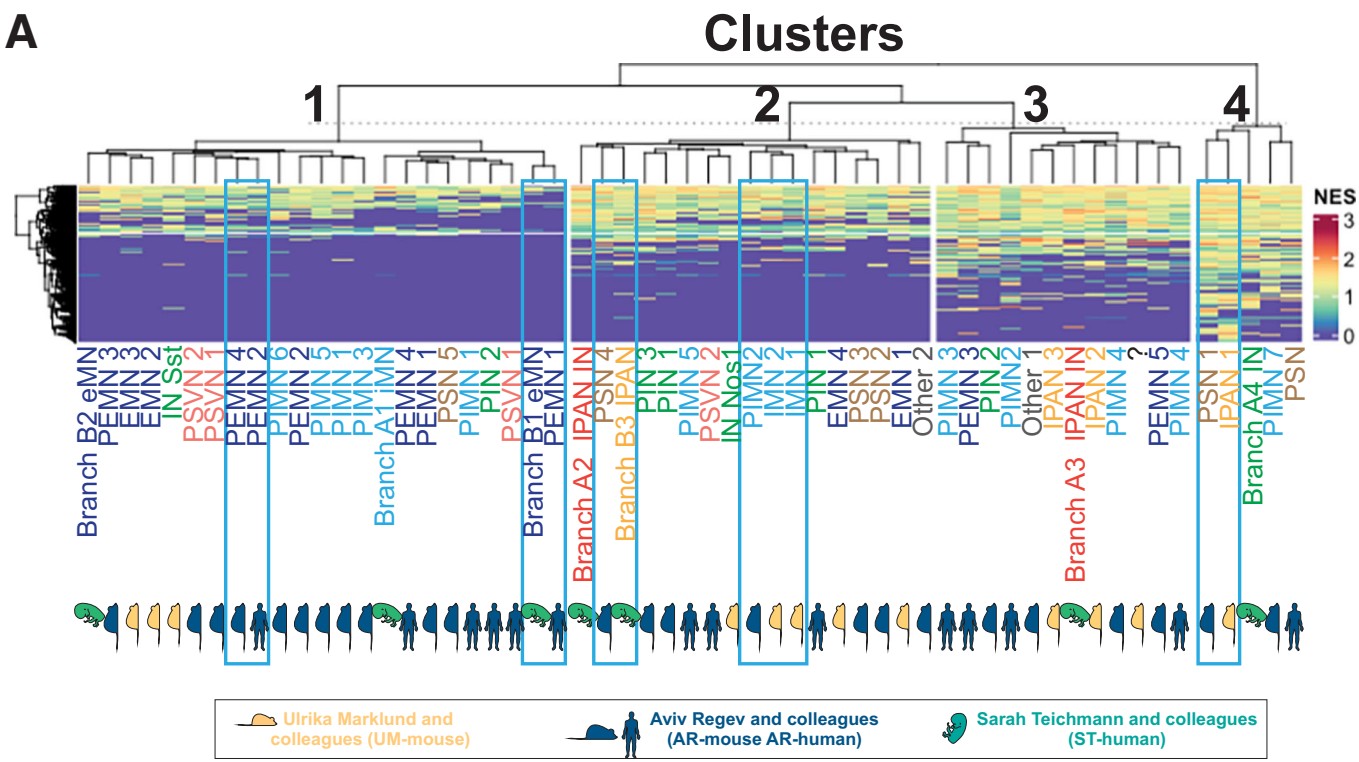

**Figure 5. Comparative analysis of primary ENS neurons using GOBP hierarchical clustering.**

Hierarchical clustering of primary enteric neuron clusters based on normalized enrichment scores of biological process gene ontology (GOBP) pathways. **(A)** Four main clusters each contain multiple enteric neuron clusters from different datasets. Blue boxes indicate closely clustered neuronal subtypes with matching functional annotation from two different datasets.

hierarchical clustering across all datasets, using the normalized enrichment score of enriched GO biological process terms found in at least one cluster (Fig. 5A). This clustering revealed only six instances of close relationships among similarly annotated neuron clusters from different datasets. Notably, AR-mouse PSN 1 and 4 cluster with UM-mouse IPAN1 and ST-human Branch B3 IPAN, respectively, while the remaining instances consist of groups of EMNs or IMNs from all four datasets. However, the majority of the dendrogram shows close relationships of clusters with diverging functional annotation both across and within datasets. For instance, AR-human PIN1 clusters closely with UM-mouse EMN4, and AR-human PIMN1, PIN 2 and PSVN 1 are all predicted to be closely related. Overall, these results suggest that while many of the primary enteric motor neurons (IMNs and EMNs) exhibit close clustering, there are notable discrepancies in the overall functional clustering and/or annotation of neuronal subtypes (Figs. 5 and EV5). To identify specific

pathways, we selected the resolution level based on the four main clusters defined in the tree structure (Fig. 5A). By averaging the normalized enrichment scores, we highlighted pathways enriched in each cluster (Fig. EV5A; Dataset EV1). At this resolution, when examining pathways uniquely enriched in each cluster, we observed that clusters 1 and 2 have few distinct pathways; most pathways enriched in these clusters also show positive scores in clusters 3 and 4. However, as shown in Figs. 5A and EV5A, clusters 3 and 4 contain pathways that are more selectively enriched in these clusters but not in clusters 1 and 2. These include pathways related to gliogenesis, neuronal projection and axogenesis for cluster 3 (Fig. EV5B), and pathways related to neuronal activity and synapses for cluster 4 (Fig. EV5C). A similar analysis, performing hierarchical clustering based on all GO biological process, molecular function, cellular component and human phenotype pathways, showed similar results to the biological process pathways alone (data not shown).

The unbiased clustering based on enrichment scores of GOBP reveals a dendrogram consisting of closely related clusters with different functional annotations. This suggests the presence of functional differences derived from the transcriptome profiles that were not adequately captured during the initial cluster annotations using only a few markers. The diverse annotations within seemingly similar clusters emphasize the necessity for a more comprehensive and thorough examination of the data when annotating clusters, even when solely considering the transcriptome as a readout of cellular identity. It is, however, important to note that this analysis and these observations are constrained by the availability of well-defined pathways related to our cell types of interest. To date, many GO biological process pathways have been described for various neuronal functions, such as synaptic transmission and response to various neurotransmitters. However, the full breadth of functions and processes performed by enteric neurons has yet to be described and mechanistically dissected to allow for their inclusion in the above analysis.

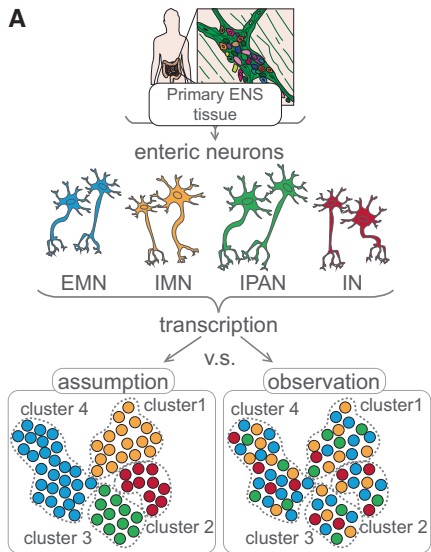

**Figure 6. Transcriptional identities are not synonymous with functional identities in enteric neurons.**

(A) Schematic illustrating the contrast between the assumed clustering of functional classes of enteric neurons in transcriptomic datasets and the organization suggested by observed transcriptional profiles.

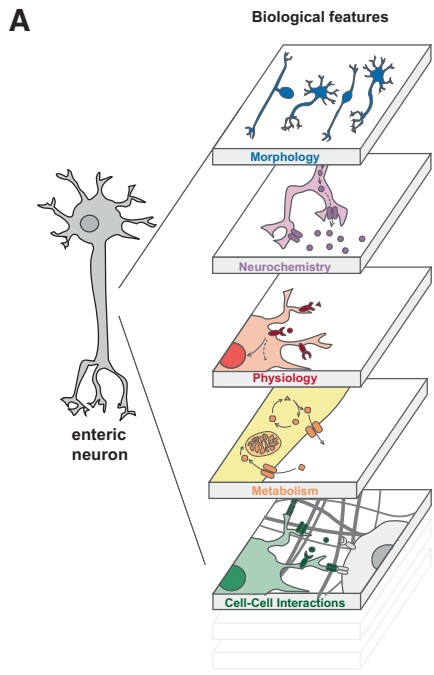

**Figure 7. Enteric neuron identity should be defined based on multiple biological features.**

(A) Multimodal characterization is essential for understanding enteric neuron function. Accurate classification requires integrating transcriptomic data with additional biological features, such as morphology, electrophysiology, metabolic and signaling profiles, and interactions with other cell types.

## Conclusions

Transcriptomic datasets of the ENS from diverse species, sexes, ages, GI regions, gut layers, and isolation methods have provided valuable resources. However, these variables introduce inherent differences that limit the generalization of findings. Paradoxically, as more ENS transcriptomic data accumulate, drawing clear conclusions becomes harder,

largely due to the absence of a unified classification system, lack of clear criteria for defining cell categories, and overreliance on transcriptomics. Because post-transcriptional regulation shapes protein output, RNA abundance only imperfectly predicts phenotype. The functional significance of transcriptional differences between neurons remains unclear and should be a research priority.

Our profiling of primary and hPSC-derived neurons revealed abundant cells with multiple neurochemical identities, consistent with prior immunohistochemical evidence for co-transmission in enteric neurons (Qu et al, 2008) and extending it to the transcriptomic scale. While immunolabeling has richly informed neurochemical coding, transcriptome-wide analyses reveal complexity not fully captured histologically. This heterogeneity has implications for ENS circuit mapping and therapeutic targeting.

Evidence suggests that ENS neurons may retain a greater degree of plasticity throughout life than the largely fixed fates of CNS neurons life (Cadwell et al, 2019; Obernier and Alvarez-Buylla, 2019; Burns, 2005; Burns and Pachnis, 2009). Neural crest-derived ENS cells have been observed to migrate, proliferate, and mature in the adult gut, potentially adapting to changes in growth, function, diet, inflammation, and injury. There is also evidence that neurons can alter transcriptional profiles and transmitter/receptor expression in response to stimuli, similar to enteric glia (Valès et al, 2018; Sanchini et al, 2023). If confirmed, such plasticity would challenge the assumption that functional differences necessarily reflect fixed lineages, and would have important implications for neuronal annotation and clustering.

scRNA-seq and snRNA-seq capture different RNA compartments, missing spatially localized transcripts crucial for neuronal function. Post-transcriptional processes—including RNA stability, editing, splicing, and translation—further shape protein diversity, much of it invisible to short-read RNA-seq. Integrating transcriptomics with translational profiling (Ribo-Tag, RiboTrap (Sanz et al, 2009), (Beach and Keene, 2008), and long-read sequencing, and accounting for PTMs, protein turnover, and localization, will better align molecular clusters with functional diversity (Fig. 6). Rigorous validation, such as the histological approach of Morarach et al (Morarach et al, 2021b), is essential.

Moving beyond transcription alone (Fig. 7) will require profiling diverse ENS samples across gut regions, models, developmental and disease states, and demographic variables (May-Zhang et al, 2021; Guyer et al, 2022). Computational tools for dataset integration and label transfer can aid subtype classification but require caution when merging heterogeneous datasets, especially from human patients. Spatial profiling, tissue clearing, and Patch-seq (Cadwell et al, 2016) can link transcriptomic identity to connectivity and physiology, though technical challenges remain.

A classification rooted in developmental ontogeny (Domcke and Shendure, 2023) may provide a more robust taxonomy, integrating lineage and molecular states across species. Lineage tracing and multi-omics can define developmental trajectories and functional diversity, but human ENS studies face tissue access and yield limitations. hPSC-derived enteric neurons (Majd et al, 2025; Fattahi et al, 2016; Barber et al, 2019) can address these gaps, enabling scalable, stage-specific cultures for genetic manipulation, high-throughput screens, and lineage tracing, complementing in vivo models.

In summary, primary ENS datasets reveal the urgent need for cautious annotation, multimodal integration, and methodological standards. Consensus on transcriptomic and functional criteria will improve understanding of ENS roles in neuropathies and DGBIs, driving more accurate cell type definitions and transformative discoveries.

## Peer review information

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

## Acknowledgements

The work was supported by DISC0-14521 and DISC2-15119 grants from the California Institute for Regenerative Medicine (CIRM), the NIH Director's New Innovator Award (DP2NS116769) and the National Institute of Diabetes and Digestive and Kidney Diseases (R01DK121169) to F.F., (F32DK121440) to R.A.G., and (R01DK119210) to A.M.G. H.M. is supported by Larry L. Hillblom Foundation postdoctoral fellowship, NIH T32-DK007418 fellowship and UCSF Program for Breakthrough Biomedical Research independent postdoctoral fellowship. We are grateful to the members of the Fattahi Lab at UCSF for their constructive feedback on the manuscript.

## Author contributions

**Homa Majd**: Conceptualization; Data curation; Formal analysis; Validation; Visualization; Methodology; Writing—original draft; Writing—review and editing. **Andrius Cesiulis**: Data curation; Formal analysis; Visualization; Methodology. **Ryan M Samuel**: Data curation; Formal analysis; Validation; Visualization; Methodology; Writing—original draft. **Mikayla N Richter**: Validation; Methodology; Writing—original draft. **Nicholas Elder**: Validation; Visualization; Methodology; Writing—original draft. **Kwun Wah Wen**: Resources. **Richard A Guyer**: Validation; Writing—original draft. **Marlene M Hao**: Writing—review and editing. **Lincon A Stamp**: Writing—review and editing. **Allan M Goldstein**: Writing—original draft; Writing—review and editing. **Faranak Fattahi**: Conceptualization; Supervision; Funding acquisition; Writing—original draft; Writing—review and editing.

## Disclosure and competing interests statement

F.F. is the inventor of several patent applications owned by UCSF, MSKCC, and Weill Cornell Medicine related to hPSC-differentiation technologies, including technologies for the derivation of enteric neurons and their application for drug discovery.

# Expanded View Figures

**Figure EV1.  Expression of cluster-specific markers across primary enteric neuron clusters.**

(A–D) UMAPs of enteric neurons generated from the original datasets of UM-mouse (**A**), AR-mouse (**B**), AR-human (**C**), and ST-human (**D**). "?" refers to the cluster labeled as "ENC11" or "?" in Morarach et al (Morarach et al, 2021b). (**E–H**) Dot plot of cluster-specific markers originally used for UM-mouse, (**E**) in UM-mouse, (**F**) in AR-mouse, (**G**) in ST-human, (**H**) in AR-human. (**I–L**) Dot plot of cluster-specific markers originally used for ST-human, (**I**) in UM-mouse, (**J**) in AR-mouse, (**K**) in ST-human, (**L**) in AR-human. (**M–P**) Dot plot of cluster-specific markers originally used for AR-mouse and AR-human, (**M**) in UM-mouse, (**N**) in AR-mouse, (**O**) in ST-human, (**P**) in AR-human. (**Q**, **R**) Bar plot proportion of different functional annotations described for (**Q**) NOS1+ and (**R**) PENK+ enteric neurons across primary ENS datasets.

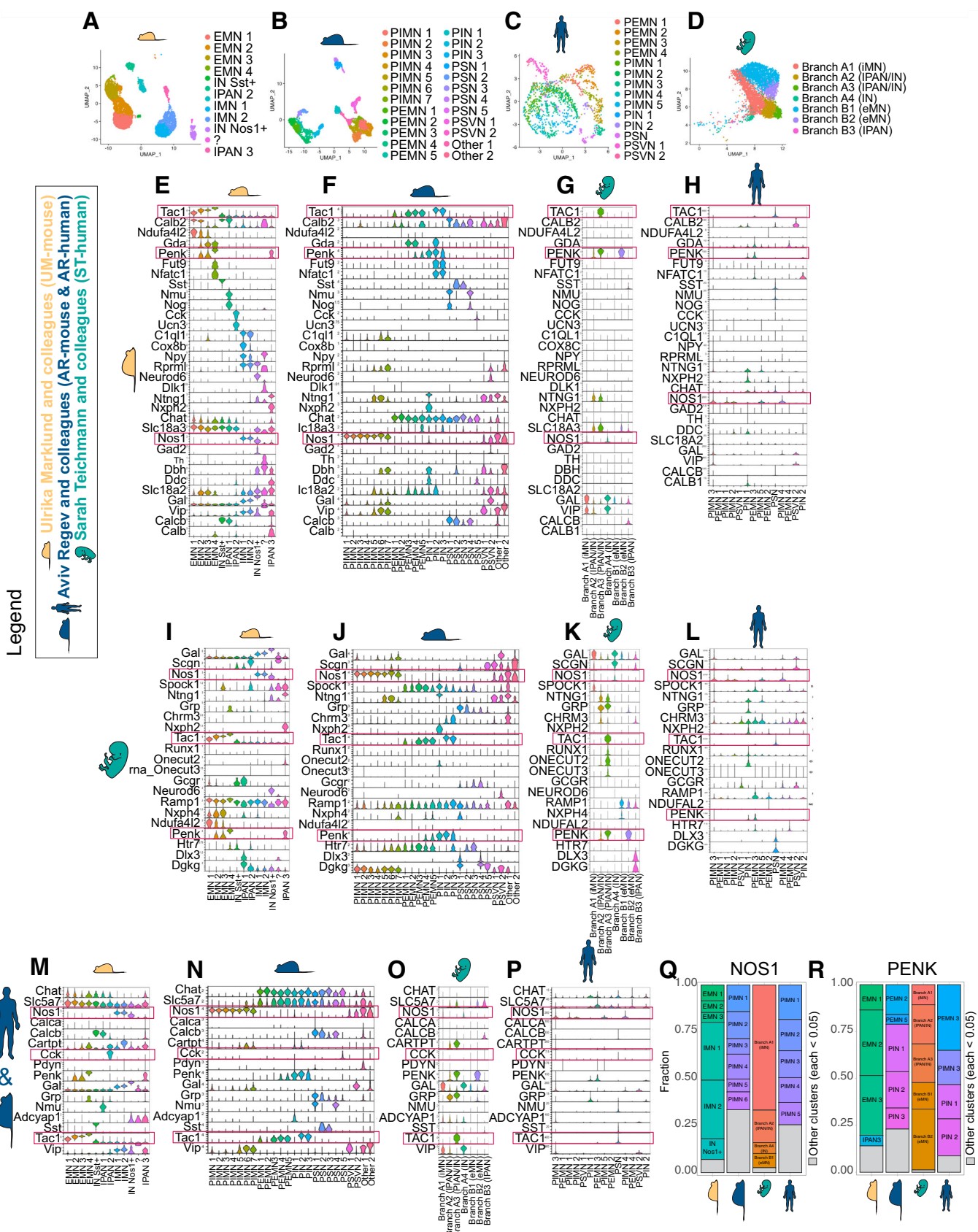

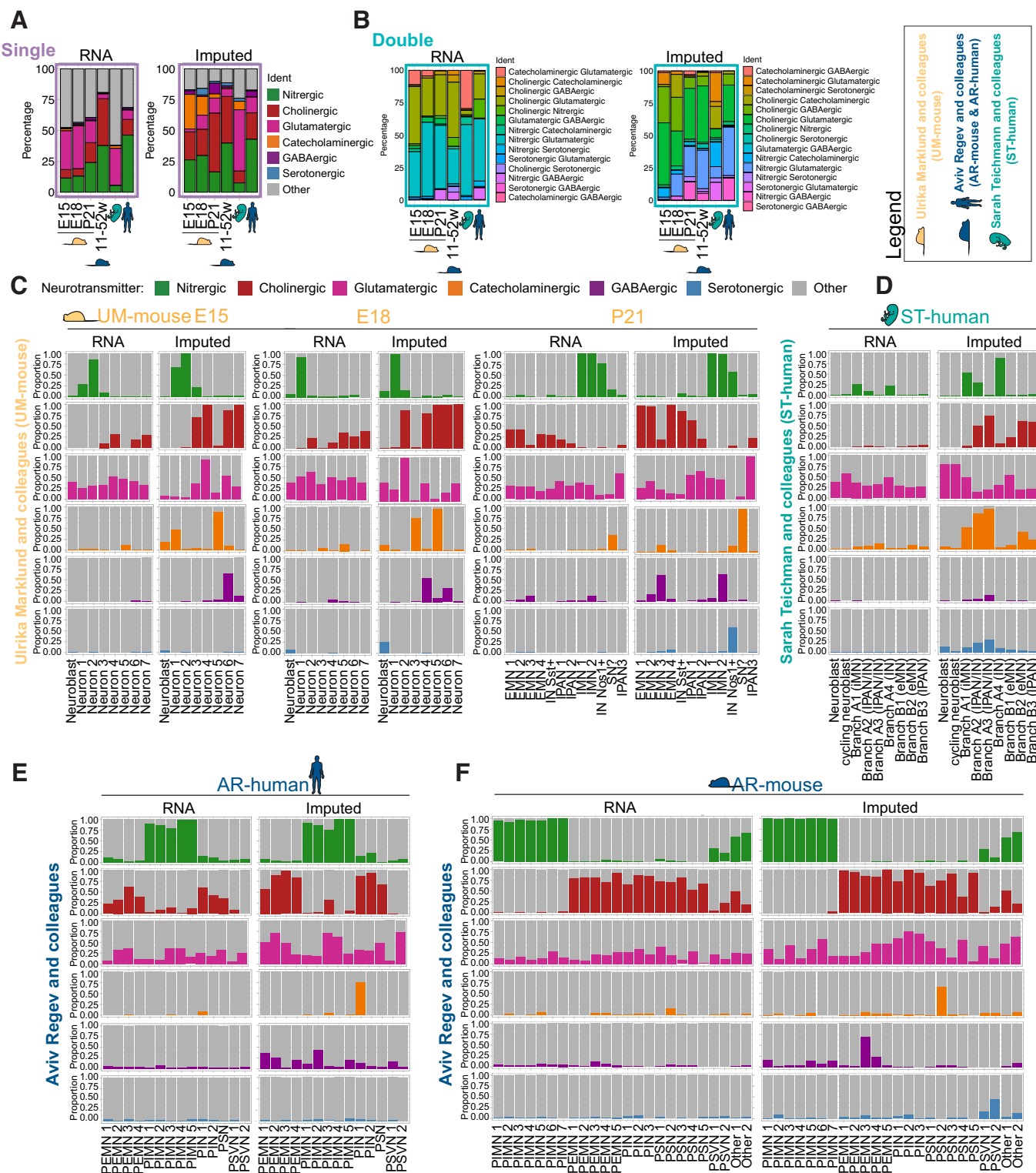

**Figure EV2. Neurochemical identities in primary enteric neuron cell types and subtypes.**

(A) Percentage of single-neurotransmitter-producing enteric neurons in primary datasets. (B) Percentage of double-neurotransmitter-producing enteric neurons in primary datasets. (C–F) Distribution of neurochemical identities in primary mouse (UM-mouse (C), AR-mouse (F)) and human (ST-human (D), AR-human (E)) enteric neuron subtypes.

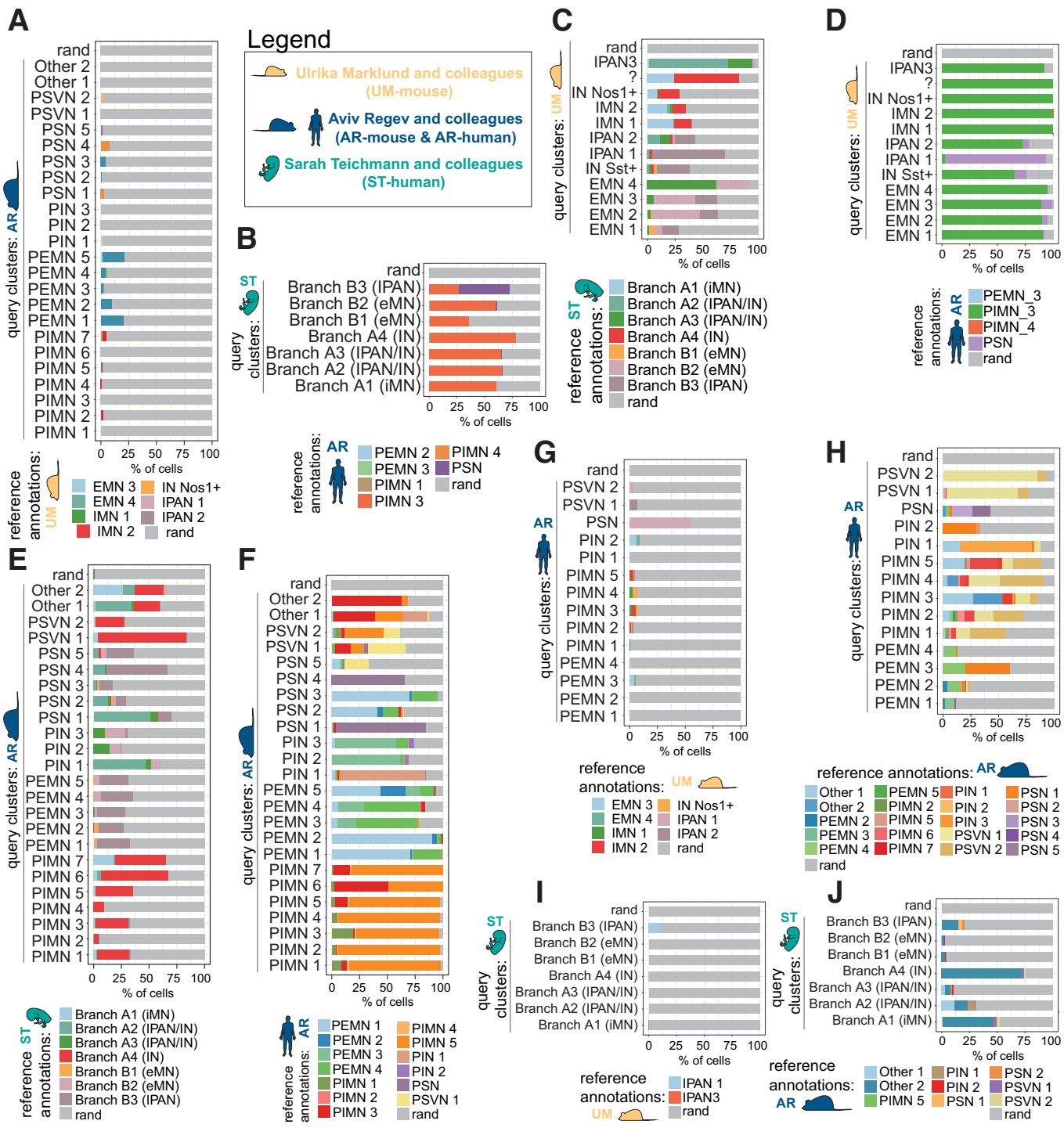

**Figure EV3. SingleCellNet unbiased label transfer and classification of primary enteric neurons.**

(A–J) Reference primary enteric neuron scRNA-seq datasets of mouse (UM-mouse (**A, G, I**), AR-mouse (**H, J**)) and human (ST-human (**C, E**), AR-human (**B, D, F**)) were used to train SingleCellNet (Tan and Cahan, 2019). These models were then used for label transfer and cross-annotation in the other datasets. Please see Methods for more details.

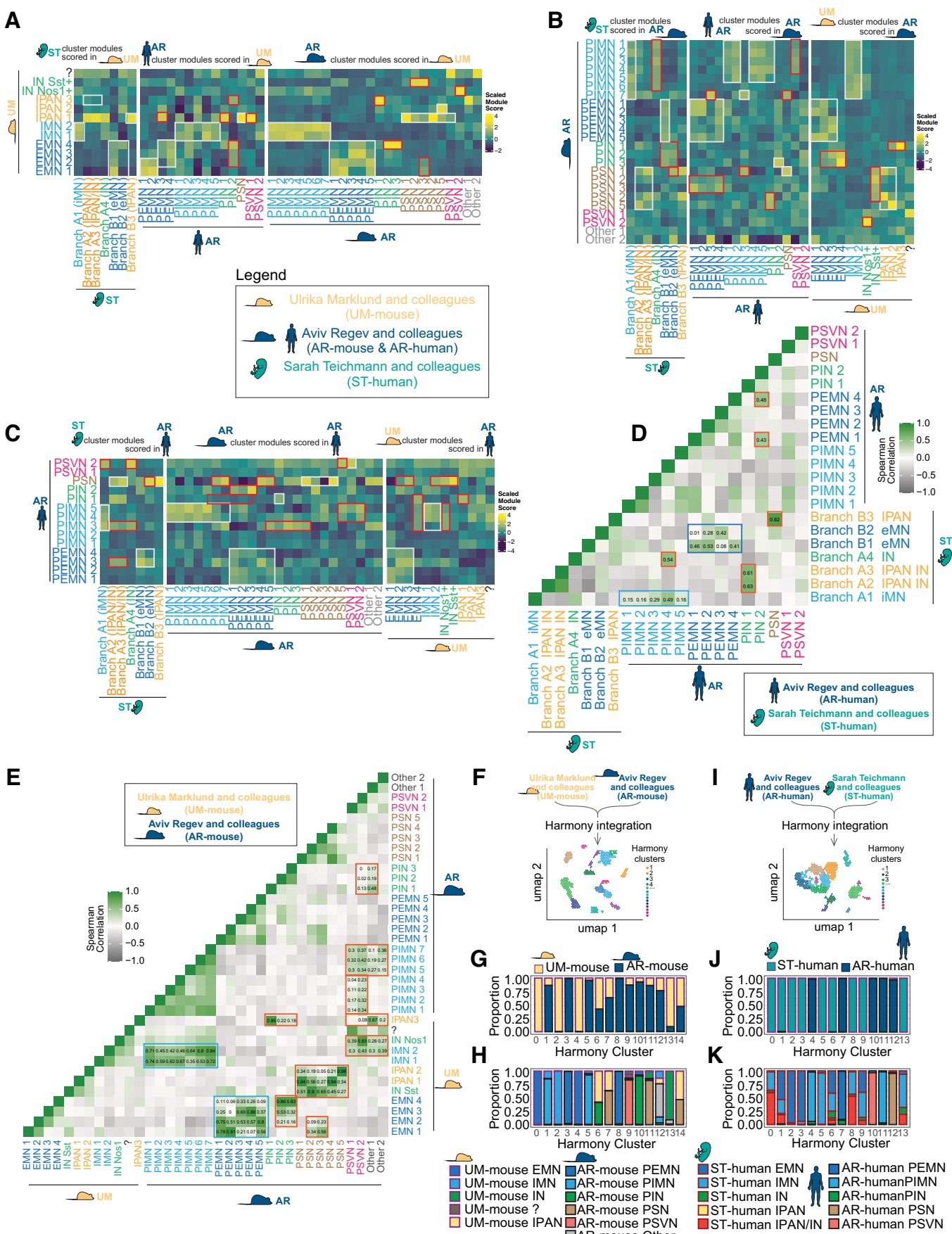

◄   **Figure EV4.   Cross-dataset module scoring, Spearman correlation of transcriptional signatures and Harmony integration of primary enteric neuron clusters.**

(A) Heatmap of the average module scores of ST-human and AR-human neuronal subtype transcriptional signatures in UM-mouse. (B) Heatmap of the ST-human, AR-human and UM-mouse neuronal subtype transcriptional signatures in AR-mouse. (C) Heatmap of the average module scores of ST-human and UM-mouse neuronal subtype transcriptional signatures in AR-human. (D, E) Heatmap matrix of Spearman correlations based on expression of 100 anchor features shared significantly variable genes (or anchor features) between (D) primary human (ST-human and AR-human) and (E) primary mouse (UM-mouse AR-mouse) enteric neuron subtypes. (F) Schematic representation of Harmony integration of UM-mouse and AR-mouse datasets. (G, H) Distribution of cells derived from UM-mouse and AR-mouse datasets (G) and their respective broad functional annotations in each Harmony cluster (H). (I) Schematic representation of Harmony integration of ST-human and AR-human datasets. (J, K) Distribution of cells derived from ST-human and AR-human datasets (J) and their respective broad functional annotations in each Harmony cluster (K). Please see Methods for more details.

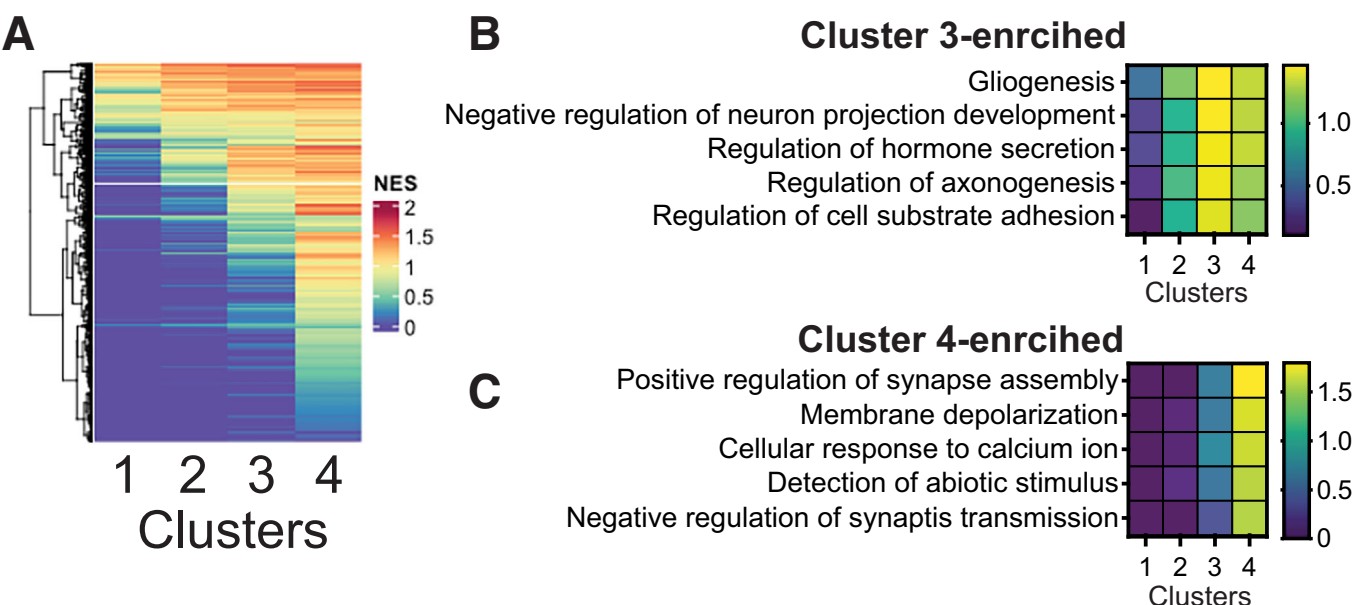

**Cluster 3-enrcihed**

**Cluster 4-enrcihed**

**Figure EV5. Hierarchical clustering and comparative analysis of primary ENS neurons.**

Hierarchical clustering of primary enteric neuron clusters based on normalized enrichment scores of biological process gene ontology (GOBP) pathways. (A) Average enrichment scores of clusters are shown in Fig. 5A. (B, C) Five representative pathways that show higher enrichment scores in clusters 3 (B) and cluster 4 (C), respectively. Please see Methods for more details.

