## [Peer Review File · The EMBO Journal]

A call for a unified and multimodal definition of cellular identity in the enteric nervous system

Homa Majd, Andrius Cesiulis, Ryan Samuel, Mikayla Richter, Nicholas Elder, Kwun Wah Wen, Richard Guyer, Marlene Hao, Lincon Stamp, Allan Goldstein, and Faranak Fattahi

Corresponding author(s): Faranak Fattahi (Faranak.Fattahi@ucsf.edu), Homa Majd (homa.majd@ucsf.edu)

Review Timeline:

Submission Date:	23rd Jan 25
Editorial Decision:	21st Feb 25
Revision Received:	28th May 25
Editorial Decision:	25th Jun 25
Revision Received:	14th Aug 25
Accepted:	18th Aug 25

Editor: Ioannis Papaioannou

Transaction Report:

Dear Faranak,

Thank you again for submitting your manuscript EMBOJ-2025-120282-T for consideration by The EMBO Journal, and for your patience during peer review. Your manuscript has now been seen by two experts in the field, and we have received their comments, which you can find below.

As you will see, referee #1 finds this piece timely, important, and useful in highlighting issues with the interpretation of single-cell sequencing data that should be taken into consideration in this field going forward. The referee also lists a number of limitations and provides detailed suggestions for the improvement of the manuscript, which would improve its clarity and increase its impact on the field. On the other hand, referee #2 is less supportive, pointing out that the different origins of the datasets re-analyzed in your study confound the interpretation of the results - with the differences potentially being attributable to either biological or technical differences.

Given the timeliness and importance of the topic, as well as the input we received from the referees, we would like to invite you to submit a revised version of your manuscript along with a detailed point-by-point response addressing all referees' comments. In line with referee #2, I should mention that the limitations of your study should be addressed either by additional analyses and/or by discussing them (as well as possible solutions) transparently and appropriately in your manuscript.

I should also add that your manuscript can only be considered further for publication in The EMBO Journal as a "Commentary" (i.e. perspective) piece, not as an original research article, since neither the scope nor the depth of the analysis would justify the latter. Therefore, I kindly ask you to take this into consideration while revising your manuscript; please focus on providing the necessary background information (also to non-specialists; please remember that The EMBO Journal has a broad readership of molecular biologists working in many different areas of biology), explaining clearly the problem/motivation for your analysis, and expanding on the conclusions highlighting the limitations and challenges, as well as your suggestions for possible solutions to the observed problems. The length of those parts of your manuscript describing data analysis should be kept to a minimum, and -again- their clarity and accessibility by our broad readership should be improved.

All the information needed to replicate your analyses should be provided either in the "Methods" section or in the legends of the respective Figures. Please retain all necessary Figures and methods in the main text of your revised manuscript; depending on the structure/extent/length of the revised manuscript, we may discuss later whether some of this information can be moved to a supplementary Appendix file or shown as Expanded View content.

Please also note that our journal encourages inclusion of data citations in the reference list to directly cite datasets that were obtained from public databases. Data citations in the article text are distinct from normal bibliographical citations and should directly link to the database records from which the data can be accessed. In the main text, data citations are formatted as follows: "Data ref: Smith et al, 2001" or "Data ref: NCBI Sequence Read Archive PRJNA342805, 2017". In the Reference list, data citations must be labeled with "[DATASET]". A data reference must provide the database name, accession number/identifiers, and a resolvable link to the landing page from which the data can be accessed at the end of the reference. Further instructions are available at: <https://www.embopress.org/page/journal/14602075/authorguide#referencesformat>.

The author contributions statement should be removed from the manuscript file. Instead, we use CRediT to specify the contributions of each author in the journal submission system. Please feel free to use the free text box to provide more detailed descriptions during submission. See also our guide to authors for more information: <https://www.embopress.org/page/journal/14602075/authorguide#authorshipguidelines>.

Please also rename the heading "Declaration of interests" to "Disclosure and competing interests statement".

When you are ready to submit your revised manuscript, please upload:

- A Word file of the manuscript text (including legends of Figures and Tables). The figures should be removed from the main manuscript file.

- Individual production-quality figure files (one file per figure). When assembling your figures, please refer to our figure preparation guidelines in order to ensure proper formatting and readability in print as well as on screen: <https://bit.ly/EMBOPressFigurePreparationGuideline>

Please note that no statistics should be calculated and shown in Figures if $n=2$. Please also note that each p value should be reported as an exact value. If the data shown in a Figure are obtained from $n \leq 2$, please use scatter plots showing the individual data points.

- i. the name of the statistical test used to generate error bars and P values (exact P values must be provided)
- ii. the number (n) of independent experiments (please specify technical or biological replicates) underlying each data point

(discussion of statistical methodology can be reported in the Methods section, but figure legends should contain a basic description of n, P, and the test applied)

iii. the nature of the bars and error bars (s.d., s.e.m.).

- A point-by-point response to the referees' comments, with a detailed description of the changes made (as a Word file). All referees' concerns must be fully addressed and their suggestions taken on board. When preparing your letter of response to the referees' comments, please bear in mind that this will form part of the Review Process File and will therefore be available online to the community. For more details on our Transparent Editorial Process, please visit our website:

<https://www.embopress.org/page/journal/14602075/authorguide#transparentprocess>

Please also check that the title and the abstract of the manuscript are brief, yet explicit, even to non-specialists. The length of the title should not exceed 100 characters, and the abstract should be a single paragraph not exceeding 175 words.

I would also like to note that all corresponding authors are required to provide an ORCID ID upon submission of a revised manuscript (<https://orcid.org/>). Please find instructions on how to link your ORCID ID to your account in our manuscript tracking system in our Author guidelines

(<https://www.embopress.org/page/journal/14602075/authorguide#authorshipguidelines>).

Further information is available in our Guide For Authors: <https://www.embopress.org/page/journal/14602075/authorguide>.

We generally allow three months as standard revision time (May 20, 2025). Should you foresee a problem in meeting this three-month deadline, please let us know in advance and we may be able to grant an extension.

Thank you for the opportunity to consider your interesting Commentary for publication in The EMBO Journal. I look forward to your revision. Please use the following link to submit it when you are ready: <https://emboj.msubmit.net/cgi-bin/main.plex>.

Best regards,

Ioannis

Referee #1:

Enteric neuroscience is currently undergoing a renaissance of sorts with renewed interest driven, in part, by multiple recent high-profile papers that used single cell RNA sequencing to gain unprecedented insight into the molecular diversity of enteric neurons and glia. However, while useful, interpreting data from these studies has often been a headache for many in the field because of issues that include discordance between data sets and apparent conflicts with known physiology. The study by Majd and colleagues does an excellent job of capturing these issues and provides a much-needed reminder to avoid overreliance on sequencing data without adequate validation. Overall, the study is timely, important, and would be useful in setting benchmarks in the field going forward. There are some minor issues that need to be addressed, but these should be easy for the authors to address.

1) Summary: I'm not sure that referring to the ENS as "rare" and "poorly understood" is the best approach here. I think this was framed a bit better in the introduction. "Rare" might cause some confusion since this is the largest division of the autonomic nervous system with hundreds of millions of cells and "poorly understood" could appear a bit dismissive of the large body of work that has been done characterizing the ENS.

2) 3rd paragraph in "Enteric neurons are diverse": Sentence 1 - It might be worthwhile to add a note here regarding how well conserved the ENS actually is between small mammals and humans. Sentence 2 - Intestinofugal neurons should be added to the list here. It also might be worth adding a bit more here to introduce the number of known subtypes (e.g. multiple populations of descending interneurons, etc).

3) Related to the point above, why do none of these datasets include an intestinofugal neuron cluster? It seems like this has

been consistently missed in every study. I wonder if the authors could use a few markers of intestinofugal neurons (e.g. Cart) to see if this is included with one of the IPAN or interneuron clusters.

4) Please double check and correct the figure legends. Several of these do not have the subpanels labeled correctly. This is pretty confusing as presented here.

5) It would be great if the authors could add a bit more description to the figure legends. This would be a huge help in understanding the significance of each panel and the various comparisons made.

6) I'm wondering if there might be a better way to represent the data shown in Figure 1 other than the multiple violin plots. The authors might consider dot plots as this could be an easier way to quickly visualize the similarities/differences.

7) I'm a bit skeptical about classifying a subset of enteric neurons as "glutamatergic" (Fig 2 and associated text). I understand that this is based on the gene expression profiles but glutamate doesn't seem to have any significant role in enteric neurotransmission. The authors should at least make a note of this in the text to help illustrate the disconnect between certain gene expression patterns and physiology.

8) If possible, it would be great to increase the size of the images shown in Figure 2 panels E and G. It would also be a good idea to adjust the color scheme to something besides red/green to make this compatible for red/green colorblind people.

9) 2nd paragraph in "Label transfer": Last sentence - IN should be interneuron not inhibitory.

10) Figure 5: Would it be possible to add a few labels to highlight key pathways in this figure? The clustering is nice but it would add a bit more meaning if you could see what these clusters relate to.

11) Conclusions: Could combining single cell approaches with a RiboTrap method get around some of the issues with RNA not always correlating with protein?

12) Conclusions (paragraph 2): I'm not sure I would agree with the author's statement "traditionally and widely accepted single-neurotransmitter classification of functional subtypes". Co-transmission is common in the ENS as it is in other branches of the ANS. I would think it would be rare for an enteric neuron type to use only a single neurotransmitter. In fact, most of the current roadmap to identifying functional types of enteric neurons relies on neurochemical coding based on co-expression of multiple transmitters. I'm also not sure I would completely agree with the statement that a "comprehensive characterization has not been carried out before". Certainly sequencing data will provide more information than immunolabeling but the available neurochemical coding of enteric neurons based on immunolabeling is pretty extensive (as far as immunolabeling goes).

13) Conclusions: How does local translation of RNA in nerve processes contribute to disconnects between snRNA-seq data and function? It might be worthwhile to add a bit of discussion to introduce this as an additional potential confounding factor.

Referee #2:

In this paper Majd et al compared four data sets containing single cell and single nuclear RNA seq data of cells within the enteric nervous system (ENS). Using a range of different bioinformatics methods, they find that the gene expression and annotation vary a lot between the data sets making it difficult to reconcile the data into a common annotation of functional subsets of neurons within ENS. The authors conclude that much more work must be done to achieve a meaningful understanding of enteric neuron entity and functions, including a variety of in situ and functional methods.

Comments:

An obvious problem with this paper is that the authors compare four data sets are from very different sources. One data set is focused on neuron development in human embryos; one is from adult human colon; one is from mouse gut; and the last is from myenteric neurons in mouse. Moreover, the data sets include RNA isolated from cells and nuclei, respectively. Although single cell transcriptomics is a very powerful method, it is well known that there are difficulties integrating data sets from different labs/papers. It is therefore not unexpected that when the authors compare data sets from different labs including cells from different species, age groups, and different locations, that the data is difficult to reconcile. Their finding could therefore represent both real differences between mice and men, age and location, and differences in methodology.

The authors propose the use of hPSC as an opportunity to study the ENS. Studies to support this notion should be discussed more in detail in the paper. They authors show results using hPSC to better annotate the cells. The data is shown without accompanying methods section. The use of immunohistochemistry I Figure 2 should also be accompanied by a detailed method description.

Conclusion:

This is basically a paper comparing four data sets of transcriptomics data from different sources and their discussion should focus on limitations/solutions for such comparisons. If the authors wanted to give a view of what is known about the ENS, they should have reviewed the literature on other aspects beyond scRNA-seq including imaging, electrophysiology, proteome, and

metabolome. They should also have discussed species differences, and differences of functions between the myenteric and submucosal plexus.

Minor:

The legends in figure 2 is wrong.

Point-by-point response file:

We are grateful to the editor and reviewers for generously dedicating their time to thoroughly review the manuscript and assist us in enhancing the quality and clarity of our study. Please find our point-by-point response to each comment below, including more detailed explanations of the revisions we have made. For easy tracking, we have used a blue font to mark the revised sections in the manuscript text and new panels in figures.

Referee #1:

Enteric neuroscience is currently undergoing a renaissance of sorts with renewed interest driven, in part, by multiple recent high-profile papers that used single cell RNA sequencing to gain unprecedented insight into the molecular diversity of enteric neurons and glia. However, while useful, interpreting data from these studies has often been a headache for many in the field because of issues that include discordance between data sets and apparent conflicts with known physiology. The study by Majd and colleagues does an excellent job of capturing these issues and provides a much-needed reminder to avoid overreliance on sequencing data without adequate validation. Overall, the study is timely, important, and would be useful in setting benchmarks in the field going forward. There are some minor issues that need to be addressed, but these should be easy for the authors to address.

We sincerely thank you for highlighting the importance and the timeliness of the study. Thank you for your thoughtful and constructive comments.

1) Summary: I'm not sure that referring to the ENS as "rare" and "poorly understood" is the best approach here. I think this was framed a bit better in the introduction. "Rare" might cause some confusion since this is the largest division of the autonomic nervous system with hundreds of millions of cells and "poorly understood" could appear a bit dismissive of the large body of work that has been done characterizing the ENS.

Thank you for your feedback. That was certainly not our intent, and we agree that describing the enteric nervous system as "rare" and "poorly understood" may unintentionally misrepresent the substantial research dedicated to it. We've revised the wording, and we believe the updated language better reflects its significance as the largest division of the autonomic nervous system (**Page 2, Summary Paragraph**).

2) 3rd paragraph in "Enteric neurons are diverse": Sentence 1 - It might be worthwhile to add a note here regarding how well conserved the ENS actually is between small mammals and humans. Sentence 2 - Intestinofugal neurons should be added to the list here. It also might be worth adding a bit more here to introduce the number of known subtypes (e.g. multiple populations of descending interneurons, etc).

Thank you for your suggestion. Please find the updated text on **Page 3, Paragraph 3**.

3) Related to the point above, why do none of these datasets include an intestinofugal neuron cluster? It seems like this has been consistently missed in every study. I wonder if the authors could use a few markers of intestinofugal neurons (e.g. Cart) to see if this is included with one of the IPAN or interneuron clusters.

Thank you for the excellent point. We have noted this in the text (**Page 4, paragraph 3**). We also checked Cart expression in these primary mouse and human enteric neuron datasets. As explained on **Page 4, paragraph 3**, and shown in **Figure S1**, Cart, expressed by gene *Cartpt* in mouse and *CARTPT* in human was detected across the datasets while none of the datasets had an intestinofugal annotation.

4) Please double check and correct the figure legends. Several of these do not have the subpanels labeled correctly. This is pretty confusing as presented here.

We are sorry about the oversight. We have double-checked and updated the figure legend labels.

5) It would be great if the authors could add a bit more description to the figure legends. This would be a huge help in understanding the significance of each panel and the various comparisons made.

As we need to keep the descriptions concise to meet editorial requirements and word limits, we have added references to the detailed descriptions in the methods section within the figure legends.

6) I'm wondering if there might be a better way to represent the data shown in Figure 1 other than the multiple violin plots. The authors might consider dot plots as this could be an easier way to quickly visualize the similarities/differences.

Thank you for your suggestion. All the **Figure 1** violin plots are replaced by corresponding dot plots as you recommended (**Figure 1 G-R**).

7) I'm a bit skeptical about classifying a subset of enteric neurons as "glutamatergic" (Fig 2 and associated text). I understand that this is based on the gene expression profiles but glutamate doesn't seem to have any significant role in enteric neurotransmission. The authors should at least make a note of this in the text to help illustrate the disconnect between certain gene expression patterns and physiology.

The inclusion of glutamatergic enteric neurons was informed by existing literature describing the identification and characterization of glutamate-containing enteric neurons, as well as glutamate responsiveness in enteric neurons expressing AMPA and NMDA receptors in both the myenteric and submucosal plexuses (Liu *et al*, 1997; Filpa *et al*, 2016). There has also been further characterization of glutamatergic neurons recently (Hamnett *et al*, 2025). However, as the reviewer noted, the functional role of glutamate signaling in this context remains

incompletely understood, and we have now highlighted this limitation in the text on **Page 5, paragraph 3**.

8) If possible, it would be great to increase the size of the images shown in Figure 2 panels E and G. It would also be a good idea to adjust the color scheme to something besides red/green to make this compatible for red/green colorblind people.

Thank you for your thoughtful note. We increased the sizes of **Figure 2E and G** panel and updated the color scheme from red/green to magenta/cyan for better accessibility.

9) 2nd paragraph in "Label transfer": Last sentence - IN should be interneuron not inhibitory.

Thank you for your attention and catching the error. The text is corrected (**Page 7, paragraph 2**).

10) Figure 5: Would it be possible to add a few labels to highlight key pathways in this figure? The clustering is nice but it would add a bit more meaning if you could see what these clusters relate to.

Thank you. Please find updated figure 5 and additional panels as well as the description of the analysis on **Page 11, paragraph 1**.

11) Conclusions: Could combining single cell approaches with a RiboTrap method get around some of the issues with RNA not always correlating with protein?

Thank you for the insightful suggestion. This is a great point and we have added it to the text on **Page 13, paragraph 1**.

12) Conclusions (paragraph 2): I'm not sure I would agree with the author's statement "traditionally and widely accepted single-neurotransmitter classification of functional subtypes". Co-transmission is common in the ENS as it is in other branches of the ANS. I would think it would be rare for an enteric neuron type to use only a single neurotransmitter. In fact, most of the current roadmap to identifying functional types of enteric neurons relies on neurochemical coding based on co-expression of multiple transmitters. I'm also not sure I would completely agree with the statement that a "comprehensive characterization has not been carried out before". Certainly sequencing data will provide more information than immunolabeling but the available neurochemical coding of enteric neurons based on immunolabeling is pretty extensive (as far as immunolabeling goes).

Thank you for your comment. We have adjusted the wording in the updated **paragraph 2 on Page 12**.

13) Conclusions: How does local translation of RNA in nerve processes contribute to disconnects between snRNA-seq data and function? It might be worthwhile to add a bit of discussion to introduce this as an additional potential confounding factor.

This is a great point, thank you. We have highlighted your point in the conclusion section (Page 12, paragraph 4).

Referee #2:

In this paper Majd et al compared four data sets containing single cell and single nuclear RNA seq data of cells within the enteric nervous system (ENS). Using a range of different bioinformatics methods, they find that the gene expression and annotation vary a lot between the data sets making it difficult to reconcile the data into a common annotation of functional subsets of neurons within ENS. The authors conclude that much more work must be done to achieve a meaningful understanding of enteric neuron entity and functions, including a variety of in situ and functional methods.

Comments:

An obvious problem with this paper is that the authors compare four data sets are from very different sources. One data set is focused on neuron development in human embryos; one is from adult human colon; one is from mouse gut; and the last is from myenteric neurons in mouse. Moreover, the data sets include RNA isolated from cells and nuclei, respectively. Although single cell transcriptomics is a very powerful method, it is well known that there are difficulties integrating data sets from different labs/papers. It is therefore not unexpected that when the authors compare data sets from different labs including cells from different species, age groups, and different locations, that the data is difficult to reconcile. Their finding could therefore represent both real differences between mice and men, age and location, and differences in methodology.

The authors propose the use of hPSC as an opportunity to study the ENS. Studies to support this notion should be discussed more in detail in the paper. They authors show results using hPSC to better annotate the cells. The data is shown without accompanying methods section. The use of immunohistochemistry | Figure 2 should also be accompanied by a detailed method description.

We appreciate the reviewer's thoughtful observation regarding the variability among the datasets we compared, including differences in species, developmental stages, gut regions, and the use of either whole-cell or nuclear RNA. Indeed, we fully agree that these differences complicate cross-study integration and interpretation, and this is precisely the central point of our discussion and conclusion. Our intention was not to gloss over these discrepancies, but rather to highlight how they challenge current efforts to define functional enteric neuron identities and subtypes using transcriptomic data alone.

This underscores our main argument: caution must be exercised when attempting to draw definitive functional conclusions based solely on transcriptional profiles especially when such

profiles are derived from heterogeneous datasets across various conditions and methodologies. The difficulty in reconciling these datasets is not a flaw in the study, but a critical insight that calls for more rigorous standards, better integrative tools, and multidimensional approaches (e.g., combining transcriptomics with electrophysiology, lineage tracing, and spatial analysis). Based on the reviewer's suggestion, we have now added a comprehensive methods section for the immunohistochemistry used in **Figure 2 (page 17, paragraph 3 and 4)**. We hope this clarifies that the observed inconsistencies among datasets are not an oversight, but rather a key takeaway we aim to communicate through this work.

Conclusion:

This is basically a paper comparing four data sets of transcriptomics data from different sources and their discussion should focus on limitations/solutions for such comparisons. If the authors wanted to give a view of what is known about the ENS, they should have reviewed the literature on other aspects beyond scRNA-seq including imaging, electrophysiology, proteome, and metabolome. They should also have discussed species differences, and differences of functions between the myenteric and submucosal plexus.

We agree that comparing transcriptomic datasets from different sources comes with significant limitations, and indeed, a central aim of our paper is to highlight these very limitations and advocate for more cautious interpretation of scRNA-seq data when used to infer enteric neuron identity and function. Our study is not intended as a comprehensive review of all aspects of ENS biology. Rather, we focus on the growing body of transcriptomic data that is increasingly used to classify enteric neurons and the challenges of interpreting such classifications in the absence of integrative, multimodal data. We emphasize throughout the manuscript that the overreliance on single-modality transcriptomics, especially when derived from diverse and non-standardized sources, can lead to premature or oversimplified functional conclusions. This is a concern we believe is both timely and necessary to voice as the field continues to generate and integrate scRNA-seq datasets. While a comprehensive review of imaging, electrophysiology, and metabolic studies is outside the scope of this focused analysis, we agree with the reviewer that such approaches are essential for advancing ENS research and to provide a fuller picture of enteric neuron identity and function.

Minor:

The legends in figure 2 is wrong.

We apologize for the error. Thank you for your attention. The legend has been corrected.

References:

Filipa V, Moro E, Protasoni M, Crema F, Frigo G & Giaroni C (2016) Role of glutamatergic neurotransmission in the enteric nervous system and brain-gut axis in health and disease. *Neuropharmacology* 111: 14–33

Hamnett R, Bendrick JL, Saha Z, Robertson K, Lewis CM, Marciano JH, Zhao ET & Kaltschmidt JA (2025) Enteric glutamatergic interneurons regulate intestinal motility. *Neuron* 113: 1019-1035.e6

Liu M-T, Rothstein JD, Gershon MD & Kirchgessner AL (1997) Glutamatergic Enteric Neurons. *J Neurosci* 17: 4764–4784

Dear Faranak,

Thank you again for submitting your revised Commentary (EMBOJ-2025-120282R) to The EMBO Journal for our consideration, and for your patience during peer review. As I have already informed you, your revised manuscript has now been seen by the original referee #1, who had previously assessed the initial version of your Commentary, and we have received their comments (included below). I am very pleased to say that the referee finds all initially raised concerns and criticisms successfully addressed, recognizes that this is an excellent work that will be of interest and useful to the community, and recommends publication without any further comments.

In light of this expert input, we are happy to move forward with your Commentary for publication in The EMBO Journal, but -as we discussed last week- there are a few changes that we would need you to make in a final version of your manuscript before we can proceed with formal acceptance and publication:

- The text should be revised/edited, throughout the manuscript, for better accessibility and readability by a broader readership of non-specialists. The revised text should be conveying in a simpler -and with fewer technical details- the motivation for this work, focusing on the conclusions and important implications of the work for the field. This information should also be better reflected in the Abstract of the revised manuscript. More technical details that might currently be detracting from the main messages and conceptual clarity should be removed from the main text, and instead included in an Appendix (please see below for more information).
- The number and complexity of main Figures should be accordingly reduced; the non-essential information or multiple panels currently included in the main Figures can be provided as either Expanded View (EV) Figures (i.e. extra figures that are presented in an expandable format inline in the HTML version of the main manuscript) and/or an Appendix (i.e. a "traditional" supplementary PDF file including any supplementary Figures, text, and tables). The heading on the first page of the Appendix PDF file should be "Appendix for" followed by the title of the manuscript, and a brief Table of Contents including page numbers for the listed items. The nomenclature for the Appendix Figures should be "Appendix Figure S#". Please make sure to also update all callouts in the main manuscript file accordingly. You can find more information on the Expanded View and Appendix options in our guide to authors: <https://www.embopress.org/page/journal/14602075/authorguide#expandedview>.
- Please note that "Figure S1" should be renamed to one of the following: "Figure #" (main Figure), "Figure EV#" (Expanded View Figure), or "Appendix Figure S#" (Appendix Figure).
- The legends of the main and EV Figures should be provided in the main manuscript file, below the References list. For the Appendix Figures, each one should be followed by its own legend in the PDF file.
- Please rename "Summary" to "Abstract".
- Please consider adding a list of up to 5 keywords after the Abstract of your revised Commentary, preferably using general/broad terms for improving search engine searchability of your manuscript.
- Please move the whole Methods section, including the two Tables, to your Appendix as "Appendix Methods". All methods and protocols should be described here in detail. Please rename the Tables within your Methods to "Table 1" and "Table 2" and make sure that they are called out in the Methods text.
- The source file name, title, legend and manuscript callout all need to be updated to "Dataset EV1" instead of "Supplementary Table 1". Its legend should be removed from the main manuscript file and uploaded as a separate tab/sheet in the same Excel file.
- Please revise the section order as follows: Title page - Abstract - Keywords - Introduction - Acknowledgements - Disclosure and Competing Interests Statement - References - Figure Legends - Expanded View Figure Legends.
- We also note that our notifications could not be delivered to the e-mail addresses of co-authors Richard Guyer and Andrius Cesiulis. Please make sure that their author profiles in our manuscript tracking system are updated with valid e-mail addresses.

Please also note that as part of the EMBO publications' Transparent Editorial Process, The EMBO Journal publishes online a Peer Review File along with each accepted manuscript. This File will be published in conjunction with your paper and will include the referee reports, your point-by-point response and all pertinent correspondence relating to the manuscript. You can opt out of this by letting the editorial office know (contact@embojournal.org). If you do opt out, the Peer Review File link will point to the following statement: "No Peer Review File is available with this article, as the authors have chosen not to make the review process public in this case."

We look forward to seeing a final version of your Commentary as soon as possible. Please let us know if you have any questions and use this link to submit your revision: <https://emboj.msubmit.net/cgi-bin/main.plex>.

Best regards,

Ioannis

Referee #1:

The authors have done a great job addressing each point raised in the initial review and have presented an even better manuscript here. In my opinion, this is excellent work that will be of high interest to the community and should serve as a benchmark for future work in this area. Congratulations to the authors on a very nice publication!

We are grateful to the editor and Reviewer 1 for generously dedicating their time to review our revised Commentary (EMBOJ-2025-120282R) for publication at The EMBO Journal. You kindly assisted us in enhancing the broader accessibility and clarity of our paper. Please find our point-by-point response to editorial comment below.

- The text should be revised/edited, throughout the manuscript, for better accessibility and readability by a broader readership of non-specialists. The revised text should be conveying in a simpler -and with fewer technical details- the motivation for this work, focusing on the conclusions and important implications of the work for the field. This information should also be better reflected in the Abstract of the revised manuscript. More technical details that might currently be detracting from the main messages and conceptual clarity should be removed from the main text, and instead included in an Appendix (please see below for more information).

Thank you for your comment. We have revised the text to reduce technicality, for example by removing details such as cluster names. The core message has been preserved, but we have simplified the examples and technical content to make the text more accessible, especially for readers outside the ENS biology field.

- The number and complexity of main Figures should be accordingly reduced; the non-essential information or multiple panels currently included in the main Figures can be provided as either Expanded View (EV) Figures (i.e. extra figures that are presented in an expandable format inline in the HTML version of the main manuscript) and/or an Appendix (i.e. a "traditional" supplementary PDF file including any supplementary Figures, text, and tables). The heading on the first page of the Appendix PDF file should be "Appendix for" followed by the title of the manuscript, and a brief Table of Contents including page numbers for the listed items. The nomenclature for the Appendix Figures should be "Appendix Figure S#". Please make sure to also update all callouts in the main manuscript file accordingly. You can find more information on the Expanded View and Appendix options in our guide to authors:

<https://www.embopress.org/page/journal/14602075/authorguide#expandedview>.

Thank you. We have substantially reduced the complexity of the main figure by moving most panels to the EV files. The appendix contains all traditional supplementary information, presented with the advised labeling format. All files have been named in accordance with the journal's preferred format.

- Please note that "Figure S1" should be renamed to one of the following: "Figure #" (main Figure), "Figure EV#" (Expanded View Figure), or "Appendix Figure S#" (Appendix Figure).

Thank you. The labels are updated.

- The legends of the main and EV Figures should be provided in the main manuscript file, below the References list. For the Appendix Figures, each one should be followed by its own legend in the PDF file.

The legends for the main and EV figures, as well as those in the appendix, have all been updated to meet the requirements.

- Please rename "Summary" to "Abstract".

Done. Thank you.

- Please consider adding a list of up to 5 keywords after the Abstract of your revised Commentary, preferably using general/broad terms for improving search engine searchability of your manuscript.

Thank you. Keywords are provided.

- Please move the whole Methods section, including the two Tables, to your Appendix as "Appendix Methods". All methods and protocols should be described here in detail. Please rename the Tables within your Methods to "Table 1" and "Table 2" and make sure that they are called out in the Methods text.

Thank you. Methods section including the tables are moved to the appendix file.

- The source file name, title, legend and manuscript callout all need to be updated to "Dataset EV1" instead of "Supplementary Table 1". Its legend should be removed from the main manuscript file and uploaded as a separate tab/sheet in the same Excel file.

Dataset EV1 is updated as requested.

- Please revise the section order as follows: Title page - Abstract - Keywords - Introduction - Acknowledgements - Disclosure and Competing Interests Statement - References - Figure Legends - Expanded View Figure Legends.

Manuscript sections are ordered as requested.

- We also note that our notifications could not be delivered to the e-mail addresses of co-authors Richard Guyer and Andrius Cesiulis. Please make sure that their author profiles in our manuscript tracking system are updated with valid e-mail addresses.

Thank you for catching this issue. The email addresses are corrected in the submission portal:

Andrew Cesiulis: Andrius.Cesiulis@ucsf.edu

Richard Guyer: rguyer2@jh.edu

Dear Faranak,

Congratulations on an excellent manuscript! I am very pleased to inform you that your Commentary has been accepted for publication in The EMBO Journal. Thank you very much for comprehensively addressing the initially raised referee concerns and the editorial requests for changes and corrections.

There is only one minor change still needed in the nomenclature of the two tables included in the Appendix file. Please rename them to "Appendix Table S1" and "Appendix Table S2", throughout the Appendix file (and include them in the Table of Contents on its title page). Additionally, please update accordingly their callouts in the main manuscript file (they must be called out, at least once each). I would be grateful if you could then please send me by e-mail the corrected Appendix PDF file and the manuscript Word file.

Your manuscript will then be processed for publication by EMBO Press. It will be copy edited and you will receive page proofs prior to publication.

IMPORTANT: Please note that you will be contacted by Springer Nature Author Services to complete licensing and payment information.

If you have any questions, please do not hesitate to contact the Editorial Office. Thank you for your contribution to The EMBO Journal. Working with you has been a pleasure!

Best regards,

Ioannis
